# The Antecedents and Consequences of Study Commitment to Online Learning at Higher Education Institutions (HEIs) in Cambodia

Chealy Chet [1], Serey Sok [2,]*[ID] and Veasna Sou [2]

[1] Rector's Office, Royal University of Phnom Penh (RUPP), Russian Federation Boulevard, Khan Toul Kork, Phnom Penh 12150, Cambodia; rector@rupp.edu.kh
[2] Research Office, Royal University of Phnom Penh (RUPP), Russian Federation Boulevard, Khan Toul Kork, Phnom Penh 12150, Cambodia; sou.veasna@rupp.edu.kh
* Correspondence: sok.serey@rupp.edu.kh; Tel.: +855-16521574

**Abstract:** A rapid spread of the COVID-19 outbreak has recently shifted teaching and learning at higher education institutions (HEIs) worldwide from the traditional classroom to the online process. During the educational disruption, online teaching and learning have become an alternative to pursue education. This paper aims to analyze the antecedents and consequences of students' study commitment at Cambodia's HEIs during the COVID-19 pandemic. The research focused upon: adaptation of online teaching and learning, consequences and antecedents of students' study commitment to online learning, and factors influencing students' willingness to participate in online learning during the post–COVID-19 pandemic. The research was primarily based on an online survey among 1002 undergraduate students at Cambodia's largest and oldest University for quantitative data. The SPSS 25 and AMOS 23 were adopted to proceed with the data analysis, especially in Confirmatory Factor Analysis (CFA) and Structural Equation Modeling (SEM). We found that: (1) two stages of online teaching and learning processes were adopted at the Royal University of Phnom Penh (RUPP): individually-managed and institutionally-managed processes; (2) the students' study commitment played an active role in improving their learning satisfaction. Meanwhile, academic support is one of the most outstanding factors influencing students' online learning; and (3) in the post–COVID pandemic, 81.4% of undergraduate students did not propose to continue online learning. The survey confirms that online learning significantly reduced their academic performance, and 62.3% claimed online teaching negatively affected their studies. A prediction reveals that gender, the effect of online learning, permanent address, and home WIFI connection influence the students' willingness to take online education in the post–COVID-19 pandemic. The findings of this research have advanced knowledge of students' study commitment and provided scientific evidence for practitioners, planners, policymakers, and researchers to promote online teaching and learning at Cambodia's HEIs during the post–COVID-19 pandemic.

**Keywords:** study commitment; online teaching and learning; higher education institutions (HEIs); Royal University of Phnom Penh (RUPP); Cambodia

## 1. Introduction

The COVID-19 pandemic has unexpectedly incurred challenges for nations worldwide. Education is one of the most impacted sectors and is now under severe threat [1]. This pandemic has negatively impacted teaching and learning globally and temporarily imposed widespread physical closures of higher education institutions (HEIs). In 2020, roughly 220 million university students were affected by the COVID-19 pandemic. The HEIs intend to shield understudies and educationalists' plans for ceaselessness of teaching and ensuring instruction fragments [2]. Therefore, policymakers, planners, practitioners, and researchers seek ways to mitigate learning losses, deploy remote learning, and reopen

the HEIs safely to solve this unexpected education disruption [3]. Online and blended learning are the choices because social distancing and staggered university entrance are measured [4]. When universities worldwide have suddenly shifted from traditional face–to-face classrooms, students from different backgrounds must take online learning. Up–to–date, the COVID-19 pandemic caused: incapacity and insufficient skills to use online application by lecturers and students [5], lack of facilities and human resources to adopt online infrastructure [6], inequities, dislocation, and instability in the educational system [7], constraints to finance and reduced income generation [8], and student mental health and psycho-social vulnerability [9,10].

However, online platforms have helped resume teaching and learning, their effectiveness and impacts have become a center of the research agenda at the HEIs across the world between 2020 and 2022. Educational administrators and researchers in different countries discussed and researched online learning in the context of rights to access education in Vietnam [11], transition experience to online learning in Saudi universities [12], efficient classrooms in China [13], students' engagement in South Africa [14], factors affecting motivation in online learning in Romania [15], and quality of life and physical activity in Brazilian universities [16]. Bryson and Andres (2020) believe that online learning adaption is a real-time experience, and the world considers online as the only alternative tool for education during the pandemic [17]. However, Wintachai et al. (2021) argue that guidelines for online teaching and learning at universities in developing and middle–income countries are insufficient [18]. For example, 69.6% of students in Dhaka of Bangladesh participated in online classes; but they felt unsafe and concerned with their mental health due to this pandemic [19]. Students prefer face-to-face learning because it is more practically interactive than online learning [20]. To increase online teaching application usage and improve the quality of education at the HEIs, Carrillo, and Flores (2020) suggested improving technological advancement for the teaching and learning process [21].

On 16 March 2020, Cambodia's Ministry of Education, Youth and Sport (MoEYS) declared a nationwide closure of 124 private and public universities in response to the global pandemic. For decades, Cambodia has focused on providing educational services in a traditional university classroom; however, the spread of the COVID-19 pandemic has needed a rapid transition to online and distance learning [22]. According to the United Nations Educational Scientific and Cultural Organization or UNESCO, the COVID-19 outbreak has placed experienced educators in a complex situation, as they are required to stay in their role. In contrast, their students must remain at home [23]. Moreover, the COVID-19 pandemic has brought unexpected and undue stress on Cambodian education stakeholders, who described increased psychosocial and mental health difficulties [24]. In addition, the pandemic may also put more pressure on the implementation of the Education Strategic Plan or ESP (2019–2023) to increase the gross enrolment rate at HEIs among populations aged between 18 and 22 years. Enrolment at HEIs did not change significantly. In the study year 2017/18, the gross enrolment rate for higher was 11.6%, which was lower than the target 23% [25].

To assist with continuous quality enhancement of distance education services during this challenging time, the MoEYS and the Education Sector Working Group (ESWG) decided to assume a comprehensive, coordinated assessment of the sector to gain evidence to help classify the best methods to alert the further expansion of COVID-19 response and recovery efforts; to assist the development of evidence-based response practices and policies, and to notify a holistic national response and recovery plan [24]. In June 2020, the MoEYS Cambodia Education COVID-19 Response Plan was also developed to effectively ensure the implementation of ESP (2019–2023) and it aims to respond to this educational crisis responsibly, effectively, and efficiently. The Ministry has also worked with key stakeholders to return lecturers and students to HEIs to teach and learn safely [26].

Online teaching and learning have become an alternative to pursue education during difficult circumstances and disruption. Investigating the antecedents and consequences of students' study commitment is essential for informing practitioners, planners, poli-

cymakers, and researchers for online teaching and learning during the post-COVID-19 pandemic. Thus, three research objectives were framed to guide this research: (1) an exploration of how online teaching and learning has been adopted and implemented during the COVID-19 pandemic; (2) a description of key consequences and antecedents of students' study commitment to online learning during COVID-19 pandemic; and (3) a prediction of factors influencing students' willingness to have online learning during the post–COVID-19 pandemic.

## 2. Conceptualizing Study Commitment of Online Learning during COVID-19 Pandemic

In this research, a conceptual framework (Figure 1 and Table A1) is used to investigate how university students commit to taking online learning at a public university in Cambodia. Study commitment of university students during the COVID-19 pandemic is either positively or negatively associated with learning capability, technical arrangement, psychological climate, faculty climate, perceived self-efficacy, academic support, and learning satisfaction. Southcombe et al. (2015) interviewed 900 academics from an Australian university and found that learning capacity enhanced study commitment [27]. A study at the Syrian Virtual University confirms a significant positive relationship between technical arrangement and study commitment [28]. At the same time, students' essential psychological climate helped to ensure their study commitment [29], and faculty climate was connected with student achievement [30]. A research result of 127 engineering students shows that self-efficacy was the semester mark predictor [31]. While Cownie (2019) raises the importance of academic support in maintaining students' study commitment [32], Ranadewa et al. (2021) reveal a positive relationship between study commitment and learning satisfaction towards online courses during the COVID-19 pandemic [33].

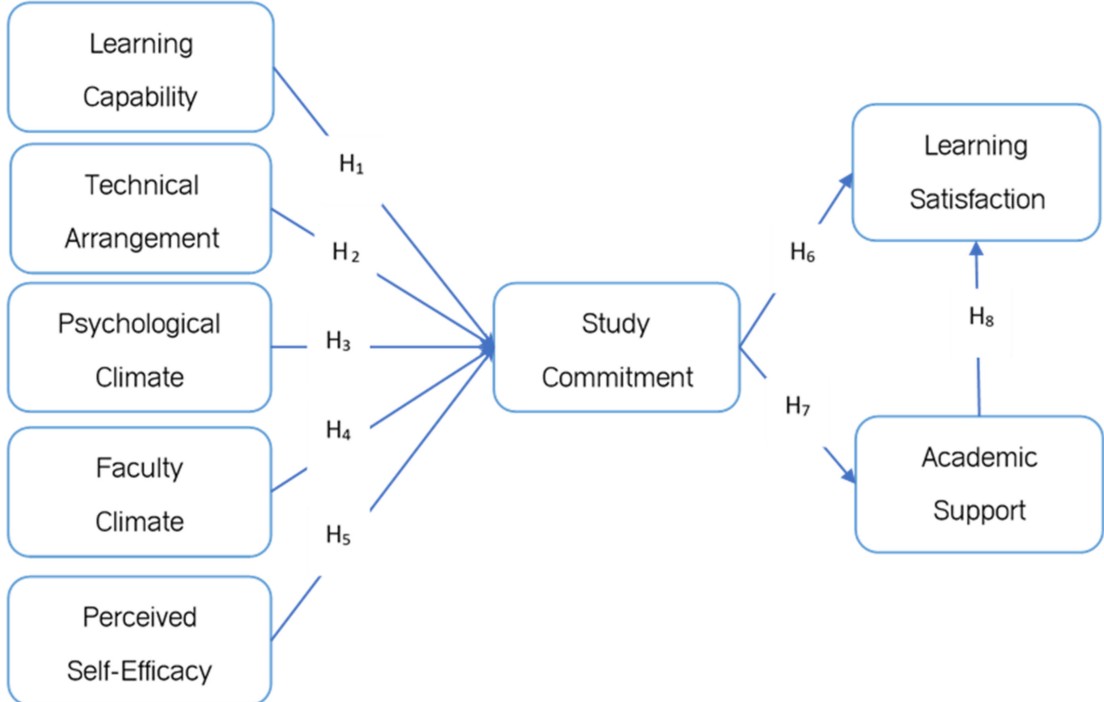

**Figure 1.** Conceptualizing students' study commitment during the COVID-19 pandemic.

As first Cambodia's modern university founded in the post–French colony, the RUPP has provided students with a face-to-face learning environment. This rapid shift into online learning has brought in a change of strategy, curriculum, and teaching approaches at HEIs. Besides the Information Communication and Technology or ICT infrastructure, financial and human resources are also required to implement online teaching and learning;

insights and perceptions of students are also necessarily investigated. Accordingly, the students' study commitment to adopt this new teaching and learning environment is the essential variable to examine the academic performance of undergraduate students during COVID-19 pandemic measures.

Based on the current literature review and actual context of online teaching and learning at Cambodia's HEIs, a hypothesis is constructed as "students' study commitment towards online learning is associated with learning capacity, technical arrangement, psychological climate, faculty climate, perceived self–efficacy, learning satisfaction, and academic support". An attempt to comprehensively analyze students' study commitment is a timely effort to advance scholarship and develop scientific evidence to promote online teaching and learning in the post–COVID-19 pandemic. However, few scholarly papers and research reports have been prepared to explore and describe the impact of COVID-19 on education [23,24,34]; the available knowledge and understanding regarding students' study commitment to online learning at HEIs are new in Cambodia. This paper not only uses Exploratory Factor Analysis (EFA), Confirmatory Factor Analysis (CFA), and Structural Equation Modeling (SEM) to test a hypothesis of students' study commitment towards online learning to discuss the core issues but also focuses on practical implications and experience to enhance the development agenda in the specific contexts of online learning and teaching. Strong scientific evidence is significant to improve Cambodia's HEIs in Cambodia in the post–COVID-19 pandemic.

## 3. Materials and Methods

The research design includes an online survey using a structured questionnaire for quantitative data and participatory approaches using an unstructured questionnaire for qualitative data. The online survey was conducted between April and November 2021 among undergraduate students at the Royal University of Phnom Penh (RUPP). After getting preliminary results, participatory and social approaches were held in December 2021 to collect quantitative data among relevant stakeholders for describing, clarifying, and explaining quantitative data. A consultative among lecturers and a group discussion among undergraduate students were also organized for qualitative data.

The RUPP is the largest public university in Cambodia. It hosts 25,106 undergraduate students, and 3482 students received full government scholarships with an enrollment rate of 18.5% in 2021. Between 1980 and 2019, 50,098 undergraduate students completed an undergraduate degree at the RUPP. The University is awarded as a full member of the ASEAN University Network (AUN). It has a unique vision "to become Cambodia's flagship university with a reputation in the region for teaching, learning, research, innovation, and social engagement." [35]. The RUPP promotes high-quality teaching and research practices to serve the community. The vision of RUPP is to be the flagship university in Cambodia, with a national standing in teaching and learning, research and innovation, and social engagement. The RUPP is currently working towards meeting the Sustainable Development Goals (SDGs), contributing at the national, regional, and global levels. The institution wishes to promote Cambodia's national cultural and natural heritage by providing high-quality research and innovation that actively engages society.

The online survey sample consisted of 1002 undergraduate students (in line with Yamane's (1967) calculations) at around a 3% confident interval [36]. The survey captured views and insights of students from all six faculties; they include Faculty of Science, Faculty of Engineering, Faculty of Social Science and Humanities, Faculty of Development Studies, Faculty of Education, and Institute of Foreign Languages (IFL). The recruited sample size represented the average study institution in their respective areas regarding the students' study commitment during online teaching and learning. For qualitative data collection, we conducted key informants with three officers from the MoEYS, a vice-rector in charge of students and ICT, four lecturers, and six undergraduate students from different faculties at the RUPP. The finding of the survey was also validated by students and lecturers. A group discussion among six students and consultative among five lecturers was organized to

present the preliminary results, collect feedback, and discuss policy application and future planning. The presentation and discussions took the form of a forum to facilitate interaction between the students, lecturers, and the researchers regarding the research findings and for purposes of validation and clarification.

The Statistical Package for the Social Science (SPSS) was used to analyze quantitative data; they included a paired-samples *t*-test for comparing the mean score of satisfaction of academic performance before and during the COVID-19 and logistic regression for exploring binary "outcome" variables, i.e., to describe key contributors to the students' wishing to take online learning in the post–COVID-19 pandemic. For testing the hypothesis of factors influencing students' study commitment towards online education, three-stage of analyses proceeded. The purpose of performing these three stages of data analysis is to double-check on reliability and validity of research variables and the meanings of questionnaire items. First, the Exploratory Factor Analysis (EFA) was performed to identify underlying research variables of learning capability, technical supports, psychological climate, faculty support, perceived self-efficacy, study commitment, learning satisfaction, and learning support. Second, the Confirmatory Factor Analysis (CFA) was used to test how well the measured variables represent the constructs and to ensure the goodness of fit for the measurement model. Third, the Structural Equation Modeling (SEM) technique with AMOS 23 was applied to explore the relationships among learning capability, technical supports, psychological climate, faculty support, perceived self–efficacy, study commitment, learning satisfaction, and learning support.

The SEM was used to predict key consequences and antecedents of study commitment among undergraduate students at the RUPP to online learning during the COVID-19 pandemic. At the same time, the CFA was also run to ensure that the model fit the SEM. The CFA and SEM models were applied to predict consequences and antecedents of study commitment among undergraduate students at RUPP. Figure 2 and Table 1 describe the second-order factor model of CFA or a measurement model. The construct validity is assessed using the guidelines of Anderson and Gerbing (1988) [37]. First, the exploratory factor analysis for all the items resulted in factor solutions, as expected theoretically. The Cronbach Alpha coefficients for each factor were greater than 0.60. Second, confirmatory factor analyses (CFA) was adopted to assess the convergent validity of the measures. The thresholds of the CFA and the SEM were adopted to evaluate the research findings of this study, such as $\chi^2$(Chi–square)/D.F (Degree of Freedom) < 3, GFI (Goodness of Fit) $\geq$ 0.90, AGFI (Adjusted Goodness of Fit) $\geq$ 0.90, NFI (Normed Fit Index) $\geq$ 0.90, CFI (Comparative Fit Index) $\geq$ 0.90, and RMSEA (Root Mean Square Error of Approximation) < 0.08, which is recommended by Anderson and Gerbing (1988); Hair et al. (2014); Jöreskog, et al. (2016); Jöreskog and Sörbom (1993); Kline (2015), and Hooper et al. (2008) [38–42].

In this research, confirmatory factory analysis consists of two main parts: (1) First Order-Factor Model (Figures A1–A8) and Second Order-Factor Mode [43] (Figure 2). This study adopted the first–order factor model to examine the research construct individually, as shown in Appendix A (i.e., Figures A1–A8) results. Each research construct has standardized loading of all items exceeding 0.60, however, excepted one research item of Q20_2, which has a score less than 0.60 (i.e., due to programing it is required to keep at least three items for each research construct), and *t*-values were higher than 1.96 (*p* < 0.001) [44]. In addition, each research construct, $\chi^2$/df was less than 2, GFI > 0.90 and AGFI > 0.90, RMR < 0.05, and *p*-value > 0.05. The second-order factor model was also adopted to examine the fitness of the overall model. All loadings exceed 0.60, and each indicator *t*-value exceeds 1.96 (*p* < 0.05), and, thus, satisfy the criteria of CFA [44,45]. The Coefficient Cronbach's $\alpha$ exceeds 0.60 for each factor. The overall fit supports the measurement model: GFI and AGFI exceed 0.9 and RMSEA less than 0.08. All these figures support the overall measurement quality given a large sample and number of indicators [46], and the measures, thus, demonstrate adequate construct validity and reliability. As shown in Table A2 (the results of Structural Equation Modeling: SEM), the overall goodness-of-fit assessment showed that $\chi^2$/df = 2.792, GFI = 0.924, AGFI = 0.907, NFI = 0.930, CFI = 0.954,

RMSEA = 0.042. The results indicated that the research model could be presented as a good fit with acceptable convergent validity. Since all values were greater than the established cutoff criteria, this study proceeds with the hypothesis testing using SEM.

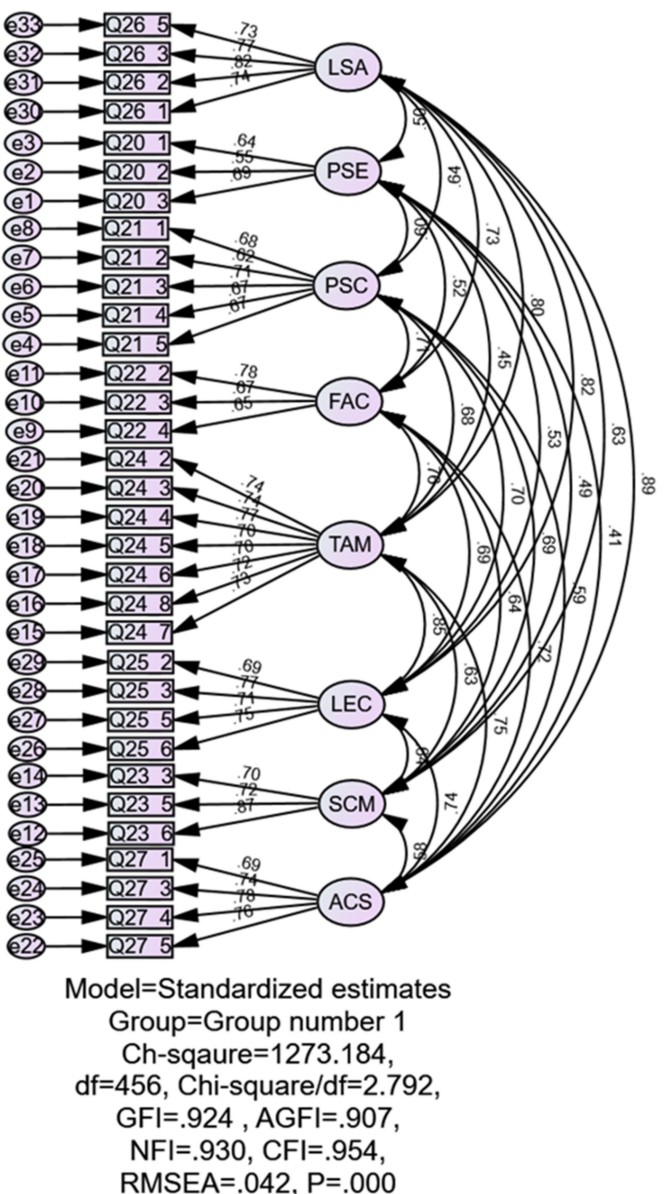

**Figure 2.** The Results of Second–Order Factor Model and Overall Model. Note: LSA: Learning Satisfaction; PSE: Perceived Self–Efficacy; PSC: Psychological Climate; FAC: Faculty Climate; TAM: Technical Arrangement; LEC: Learning Capability; SCM: Study Commitment; ACS: Academic Support.

As the data collection process for the research variables were obtained from the same source from student sides who learn at one university, there is a possibility that common method variance needs to be concerned [47,48] to confirm if the strengths of the relationships among research constructs have been inflated or deflated [49]. To assess the potential impact of this form of bias in this study, its discriminant validity is tested in two steps. First, a Harmon one-factor test is adopted [48] that loads all the variables into a principal component factor analysis. This procedure means that if the newly introduced common latent factor explains more than 50% of the variance, common method bias may be present [50]. The results reveal that a solution accounts for 36.475% of the total variance, which is less than 50% as recommended by Podsakoff, MacKenzie, and Podsakoff (2012) [51] (also refer to Jakobsen and Jensen, 2015) [52] (the result of CMV is referred

to Table A3). Second, convergent validity was demonstrated, as the average variance extracted (AVE) values for all constructs were higher than the suggested student value of 0.50 [53]. However, the AVE of research variables of "Perceived Self–Efficacy" has 0.394, "Psychological Climate" has 0.387, and "Faculty Climate" has 0.498, which is lower than the cut-off value. This process is still keeping those research items because when we decided to delete some of them, the results affected other research items. Thus, discriminant validity was determined by comparing the square root of the AVE with the Pearson correlations among the constructs. All AVE estimates from Table 2 (Second–Order Factor Model of CFA) are greater than the corresponding inter-construct square correlation estimates in Table 1 (Correlation Matrix). Based on these results, it seems that the common method bias is unlikely to be a problem with the data [54,55].

**Table 1.** The Results of Correlation Matrix (n = 1002).

| Variables | Mean | Std. D | 1 | 2 | 3 | 4 | 5 | 6 | 7 | 8 |
|---|---|---|---|---|---|---|---|---|---|---|
| **1–OLE** | 3.752 | 0.712 | 1.00 | 0.426 ** (0.181) | 0.318 ** (0.101) | 0.376 ** (0.286) | 0.345 ** (0.120) | 0.297 ** (0.09) | 0.385 ** (0.148) | 0.373 ** (0.139) |
| **2–PEC** | 3.026 | 0.876 | | 1.00 | 0.529 ** (0.279) | 0.599 ** (0.358) | 0.575 ** (0.331) | 0.476 ** (0.226) | 0.560 ** (0.314) | 0.513 ** (0.263) |
| **3–FAC** | 2.845 | 0.935 | | | 1.00 | 0.488 ** (0.238) | 0.598 ** (0.358) | 0.550 ** (0.302) | 0.531 ** (0.282) | 0.573 ** (0.328) |
| **4–AIC** | 2.912 | 1.009 | | | | 1.00 | 0.518 ** (0.268) | 0.475 ** (0.226) | 0.514 ** (0.264) | 0.512 ** (0.262) |
| **5–TRL** | 2.724 | 0.858 | | | | | 1.00 | 0.634 ** (0.402) | 0.723 ** (0.522) | 0.681 ** (0.464) |
| **6–ISP** | 2.450 | 0.930 | | | | | | 1.00 | 0.611 ** (0.373) | 0.729 ** (0.531) |
| **7–LEC** | 2.879 | 0.923 | | | | | | | 1.00 | 0.683 ** (0.466) |
| **8–OLP** | 2.869 | 0.962 | | | | | | | | 1.00 |

Note: **: Correlation is significant at the 0.01 level (2–tailed). A 5-point scale was measured, and all values in **parentheses and bold** are square correlations between two research constructs. The Pearson Correlation method was adopted. 1 = OLE: Online Learning Performance; 2 = PEC: Personal Concerns; 3 = FAC: Faculty Concerns; 4 = AIC: Academic Impact Concerns; 5 = TRL: Transition to Remote Learning; 6 = ISP: Impact on Study Performance; 7 = LEC: Learning Challenge; 8 = OLP: Online Learning Performance.

The Average Variance Extracted (AVE) and Composite Reliability coefficients (CR) were applied to relate the quality of a measure. To avoid misconceptions, it is needed to appropriately understand the equations of the AVE and CR, as well as their association to the definition of validity and reliability. In this manuscript, we explain, using simulated one-factor models, how the number of items and the homogeneity of factor loadings might influence the AVE and CR results.

$$\text{AVE} = \frac{\sum_{i=1}^{n} \lambda_i^2}{n} \tag{1}$$

$$\text{CR} = \frac{\left(\sum_{i=1}^{n} \lambda_i\right)^2}{\left(\sum_{i=1}^{n} \lambda_i\right)^2 + \left(\sum_{i=1}^{n} \delta_i\right)} \tag{2}$$

where: $\lambda$ (Lamda) represents the standardized factor loading, and $i$ is the number of items (1) and $\delta$ (Delta) represents error variance terms (2) while $\delta = 1 - \lambda_i^2$.

According to Fornell and Larcker (1981b) [56] and Peterson and Kim (2013) [57], AVE must exceed 0.50, and CR must exceed 0.70, respectively. J. F. Hair, Black, Babin, and Anderson (2014) recommend that the *t*-value must be greater than 1.96 and the *p*-value < 0.05 [38].

**Table 2.** The Results of Confirmatory Factor Analysis (CFA): Second–Order Factor Model.

| Indicators | | Research Constructs | Standardized Loading > 0.60 | *t*-Value > 1.96 | AVE > 0.50 | CR > 0.70 |
|---|---|---|---|---|---|---|
| **Q20_3** | ← | **Perceived Self-Efficacy** | 0.685 | A | | |
| Q20_2 | ← | | 0.546 | 13.156 | 0.394 | 0.66 |
| Q20_1 | ← | | 0.645 | 14.161 | | |
| Q21_5 | ← | **Psychological Climate** | 0.675 | 19.634 | | |
| Q21_4 | ← | | 0.672 | 19.244 | | |
| Q21_3 | ← | | 0.711 | A | 0.387 | 0.73 |
| Q21_2 | ← | | 0.622 | 17.81 | | |
| Q21_1 | ← | | 0.685 | 19.726 | | |
| Q22_4 | ← | **Faculty Climate** | 0.653 | A | | |
| Q22_3 | ← | | 0.674 | 21.684 | 0.498 | 0.75 |
| Q22_2 | ← | | 0.784 | 18.788 | | |
| Q23_6 | ← | **Study Commitment** | 0.874 | A | | |
| Q23_5 | ← | | 0.716 | 23.735 | 0.587 | 0.81 |
| Q23_3 | ← | | 0.696 | 22.657 | | |
| Q24_7 | ← | **Technical Arrangement** | 0.732 | 23.958 | | |
| Q24_8 | ← | | 0.724 | 23.421 | | |
| Q24_6 | ← | | 0.705 | 22.719 | | |
| Q24_5 | ← | | 0.703 | 29.663 | 0.539 | 0.63 |
| Q24_4 | ← | | 0.769 | A | | |
| Q24_3 | ← | | 0.735 | 23.753 | | |
| Q24_2 | ← | | 0.737 | 23.795 | | |
| Q27_5 | ← | **Academic Support** | 0.76 | 24.393 | | |
| Q27_4 | ← | | 0.777 | A | 0.552 | 0.74 |
| Q27_3 | ← | | 0.745 | 23.824 | | |
| Q27_1 | ← | | 0.686 | 21.713 | | |
| Q25_6 | ← | **Learning Capability** | 0.75 | 23.766 | | |
| Q25_5 | ← | | 0.705 | 22.2 | 0.530 | 0.72 |
| Q25_3 | ← | | 0.769 | A | | |
| Q25_2 | ← | | 0.686 | 21.553 | | |
| Q26_1 | ← | **Learning Satisfaction** | 0.745 | 31.057 | | |
| Q26_2 | ← | | 0.818 | A | 0.589 | 0.77 |
| Q26_3 | ← | | 0.773 | 26.738 | | |
| Q26_5 | ← | | 0.732 | 24.953 | | |
| **Goodness-of-Fit Index (The Results)** | | | **Goodness-of-Fit Index (The Threshold Value)** | | | |
| $\chi^2$/D.F = 2.792 | | | $\chi^2$/D.F < 3 | | | |
| GFI = 0.924 | | | GFI $\geq$ 0.90 | | | |
| AGFI = 0.907 | | | AGFI $\geq$ 0.90 | | | |
| NFI = 0.930 | | | NFI $\geq$ 0.90 | | | |
| CFI = 0.954 | | | CFI $\geq$ 0.90 | | | |
| RMSEA = 0.042 | | | RMSEA < 0.08 | | | |

Note: A = parameter regression weight was fixed at 1.000 and significant *p*-value < 0.05 and *t*-value > 1.96.

## 4. Results

*4.1. Online Learning and Academic Performance during COVID-19 Pandemic*

Since its establishment in 1960, teaching and learning at the RUPP have been designed to deliver lectures in classrooms. The COVID-19 pandemic shocked a sharp policy change of Cambodia's HEIs. An immediate replacement of online learning and teaching has brought the RUPP constraints, new experience, and a great move from traditional education towards an adaptation of digital education. After getting instruction from the MoEYS on 13 March 2020, all the HEIs in Cambodia started to operate online teaching and learning with limited experience, ICT infrastructure, and resources. Figures 3 and 4 describe different types of electronic devices and platforms students use at the RUPP during online learning. While smartphone (86.6%) was the most used electronic device, Microsoft Teams became the highest used platform for students. Platforms such as Zoom (33.0%), Google Meet (24.7), Messenger (9.8%), and Skype (1.1%) were the alternative video conferencing and social media. The survey shows that few students borrowed smartphones from their family members or friends; they did not own this device for learning online. However, laptops were quite convenient for learning; only 58.1% of the students used them for online learning compared to their ownership of 68.7%.

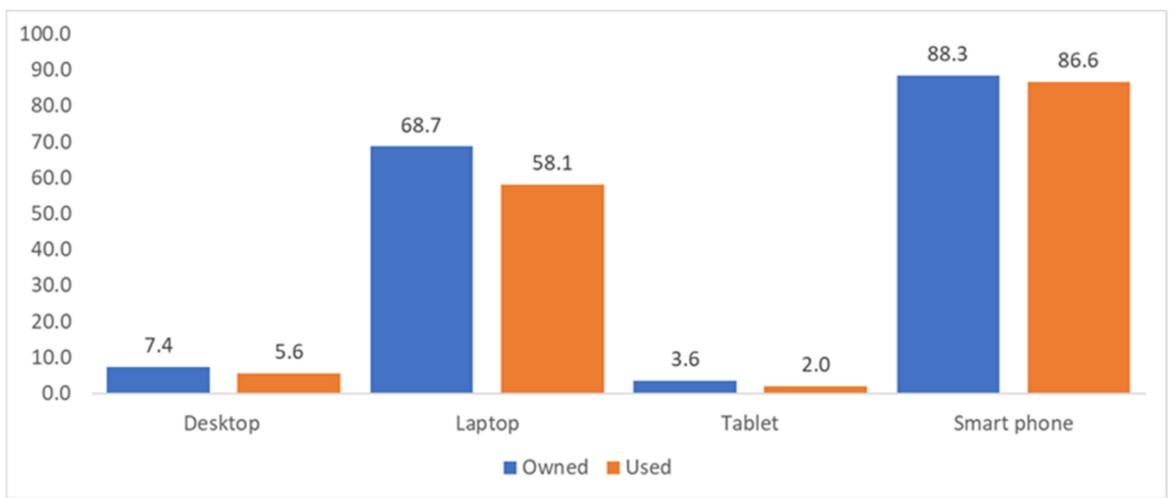

**Figure 3.** Electronic devices owned and used by students for online learning (n = 1002).

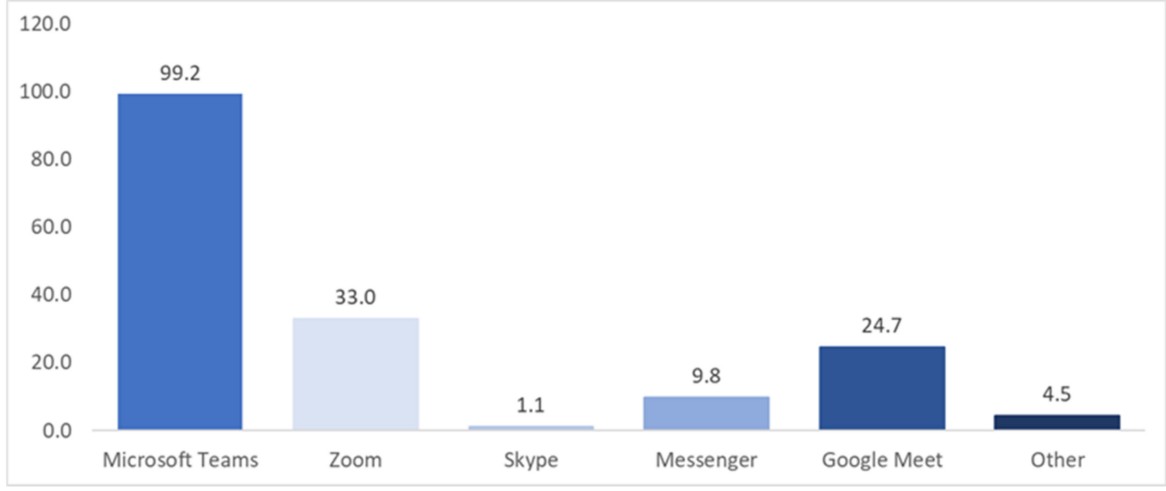

**Figure 4.** Platforms owned and used by students for online learning (n = 1002).

At RUPP, classes resumed during the COVID-19 pandemic through two stages: individually-managed and institutionally-managed online teaching and learning process. At first, lecturers continued their classes by individually-managed online teaching and learning process when the MoEYS issued a letter on 13 March 2020 to public HEIs to offer distance learning to replace physical classes [24]. Social media and video conferencing platforms played a significant role in operating online teaching and learning. Lecturers freely used their available social media and video conferencing platforms such as Skype, Messenger, Telegrams, and Facebook to resume their classes. Most lecturers and students had already experienced using Skype, Messenger, Facebook, and Telegram for teaching and communication. Other platforms such as Zoom, Microsoft Teams, Google Meet, and Google Form were then not so familiar in Cambodia.

When the world learned that the COVID-19 pandemic would not end soon, the RUPP established the online teaching and learning support committee to provide technical assistance for lecturers, students, and staff. In addition, the RUPP shifted to carry out institutionally-managed online learning and teaching processes in November 2020. The Vice-Rector in charge of students and ICT describes the RUPP pivoted physical classes into online learning with dilemma approaches. The RUPP established a committee to manage and support online teaching and learning. The committee has primary task support lecturers and students access Microsoft Teams. Therefore, the Information Technology or IT team was working to establish e-mail accounts of students and lecturers to enable them to use licensed Microsoft Teams. During the COVID-19 pandemic, the IT team has created 37,960 accounts for students and 1809 accounts for lecturers and staff; they were an enormous burden on the IT team. For exams, hybrid assessments were applied; some faculties organized assessments based on Standard Operation Procedure (SOP) issued by MoEYS. Exam takers in physical classes were broken down into small groups; only selected subjects were assigned for exams. Students' portfolios and folders were required and collected, and online assessments were partially done by the feasible department. The committee also conducted real-time monitoring to support students while taking classes and examinations [Pers. Comm. KII–4].

However, the RUPP did not plan for unexpected disruptions of physical classes; the University could adopt online teaching and learning because of the available hardware and software. The RUPP adopted Microsoft Teams as an institutionally-managed online teaching and learning platform because Microsoft Cambodia provided an unlimited Office 365 with the online version in March 2019. At the same time, the first smart classroom in Cambodia was established and soft-launching in February 2020 with the cooperation between RUPP and the International Centre for Higher Education Innovation under the auspices of UNESCO (UNESCO–ICHEI). This smart classroom has several functions such as real-time broadcasting teaching, inter-activities of teachers and students by interactive all-in-one teaching machine and learning management system (LMS), online and blended learning, and digital course development. The University also established ten video conference rooms to assist lecturers who require necessary facilities and techniques for online teaching. According to the Vice-Rector, the ICT software, hardware, and capacity building of Office 365 applications have gradually developed for blended learning start-up since early 2019; they then became essential resources to operate the online teaching and learning at the RUPP during the COVID-19 pandemic [Pers. Comm. KII–4].

In early 2019, the RUPP worked with Microsoft Cambodia and UNESCO–ICHEI to develop the ICT software and hardware to start blended learning. Microsoft Cambodia has offered free licenses of Office 365 (online version) with an account of two terabytes to each student, lecturer, and staff, helping the RUPP to operate blending learning. Moreover, the RUPP builds the capacity of lecturers and students to use Office 365 because the available online productivity apps (i.e., Word, Excel, PowerPoint, OneNote, OneDrive, Outlook, and Teams) are beneficial for educational purposes. Microsoft Cambodia is also provided with affordable Office 365 with an offline version at 22 US dollars per lecturer or staff; it costs up to 170 US dollars per person in the regular market. Shortly, the RUPP

is planning to purchase the offline version for lecturers and staff to promote teaching and learning. Actually, the RUPP had already established some of the ICT infrastructure, building capacity, and platform for online teaching and learning. Therefore, the RUPP could provide online education without difficulties during the COVID-19 pandemic [Pers. Comm. KII–4].

The survey shows that 82.3% of the students at the RUPP were from 25 different provinces and cities of the country. Out of the total, only 33.7% of the students had access to WIFI connection at their home; they mainly resided in Phnom Penh and provincial towns. During the group discussion, students described how they dealt with their online learning. Almost all of the students returned to their hometowns, especially after the community outbreak on 20 February 2020. Students agreed that smartphones were beneficial for their online learning. The majority of the students were familiar with social media and video conferencing applications, but they had difficulties accessing the electronic devices. Both lecturers and students used their own electronic devices and internet connection. All combined costs were hardly covered by students from different provinces. Most of the students could not afford both smartphones and laptops; they could buy only one of them. A sophomore at the Faculty of Social Science and Humanity [Pers. Comm. KII–6] describes that:

> *I use a smartphone for online learning because I can connect to the internet at home. It is tough for me to follow the lecturing session because internet service is very poor in my area [Banteay Meanchey Province]. I can only afford to buy either a smartphone or a laptop. If I buy a laptop, I cannot access the internet because my home is not connected to WIFI. Now I struggle with various challenges, mainly document downloads and software installation. I am taking a statistics class, and I cannot practice Statistical Package for the Social Sciences (SPSS). My lecturer is not so happy because I do not practice. I cannot also borrow a computer from my classmate for practice or do the assignment because I am recently in my home town. My lecturers said that I need to have a laptop for practicing, doing my mid-term paper, and my final examination. All those tasks require me to run SPSS software.*

The consultative meeting among lecturers confirms that online teaching started with difficulty initially; it has now become normalized. A lecturer raised that "Yes, it is not about a choice; but it is the only alternative way to continue teaching. We [lecturers] do not want to fear getting COVID-19 while we require to teach the students. At the same time, we also do not want to stop teaching. So online learning is the best option. At least, lecturers can pursue non-lab studying activities." Lecturers also raised some issues they and students faced; their challenges included pedagogical problems, internet connection, and some other ICT skills. The majority of students were from different provinces; they did not have sufficient resources for online learning at all [Pers. Comm. CM–1]. An official from the Department of Planning at MoEYS mentioned that online learning was not a new idea to Cambodia. So far, there have been various online learning platforms, yet some lecturers and students were not aware of or experienced in them. Unfamiliarity to those platforms and poor internet connection was a struggle and shocked them while the way of learning was transformed during the pandemic. In contrast, the pandemic can be an opportunity to modernize university learners to adapt to the digital learning context. It is essential to enhance their skills more competitively and open up their world [Pers. Comm. KII–1].

*4.2. Key Consequences and Antecedents of Students' Study Commitment to Online Learning during COVID-19 Pandemic*

The CFA, which used the same variables as illustrated in Table 2 (Second–Order Factor Model of CFA), was run before proceeding with the SEM to test the likelihood estimation method. The second-order factor model is accepted to test the overall research variables [37]. The results show that goodness–of–fit measurements were acceptable (i.e., GFI = 0.926, AGFI = 0.908, NFI = 0.928, CFI = 0.951, and RMSEA = 0.044) (Figure 5); this indicates that the proposed model is satisfactory with goodness-of-fit assessment [44].

The SEM model reveals (Table A2 and Figure 5—the result of SEM): learning capability ($\beta$ = 0.39 ***; $p < 0.001$; *t*-value = 6.33), technical arrangement ($\beta$ = 0.31 ***; $p < 0.001$; *t*-value = 5.126); psychological climate ($\beta$ = 0.15 **; $p < 0.05$; *t*-value = 3.093), faculty climate ($\beta$ = 0.24 ***; $p <0.001$; *t*-value = 5.797); and perceived self-efficacy ($\beta$ = 0.14 ***; $p < 0.001$; *t*-value = 3.990) have a positive significant impact on study commitment, respectively. The model also predicts that study commitment has a significant and positive influence on learning satisfaction ($\beta$ = 0.81 ***; $p < 0.001$; *t*-value = 17.33) and academic support ($\beta$ = 0.29 ***; $p < 0.001$; *t*-value = 5.296). In the same time, academic support has a significant and positive influence on learning satisfaction ($\beta$ = 0.62 ***; $p < 0.001$; *t*-value = 10.54). The SEM model indicates that the students' study commitment played a vital role in enhancing their learning satisfaction. Indeed, academic support was one of the most influencing factors increasing student learning during their online studies. The same model also predicts that study commitment is a key mediating variable to associate between independent variables (i.e., learning capability, technical arrangement, psychological climate, faculty climate, and perceived self-efficacy) and dependent variables (i.e., learning satisfaction and academic support). The mediating effects of students' study commitment among research variables are shown in Table A4 (the results of mediating effects by Sobel's test). There are many statistical methods to test mediation effects, such as hierarchical regression [58] and SEM. Thus, the Sobel test was adopted for this study. The Sobel's statistical procedure test involves two phases. First, there is a significant mediated effect if the *z*-test exceeds *t*-value = 1.96 for 2-tailed tests with an $\alpha$ = 0.05 [59–61].

According to Hair et al. (2010), and MacKinnon, Warsi, and Dwyer (1995), the indirect effect was calculated using the following formula: indirect effect = a × b (where a is the path coefficient of the relationship between the independent and the mediator variables, and b is the path coefficient of the relationship between the mediator and the dependent variables) [44,62]. Second, the significance level of the *z*-test was computed using the Sobel test, as follows:

$z$-test = $\frac{ab}{\sqrt{b^2 SE_a^2 + a^2 SE_b^2}}$, where $SE_a$ is the standard error (SE) of the relationship between the independent and the mediator variables, and $SE_b$ is the standard error (SE) of the relationship between the mediator and the dependent variables (see [60]). Therefore, the results of mediating effects in this study are shown in Table A4 (the results of mediating effects by Sobel's test) and Figures A9–A18 showed the suggested model for mediating effect's procedure, respectively (cf., [63]).

The result of group discussion among students reveals that learning online required high commitment; they described both satisfaction and dissatisfaction with this new approach. The student commitment depended on their patience, confidence, and adaptation to the new learning environment. Some students struggled to overcome challenges and constraints to fulfill course requirements successfully [Pers. Comm. FDG–1]. In contrast, students could take online classes at any place and time. Some students felt more focused on taking online courses because they could study in their own spaces and have fewer distractions. Online learning was less of a hassle because students did not feel crowded in a classroom of more than 30 students. Moreover, students could save time traveling to schools. Furthermore, students could review the video recorders if they missed the classes. Students could pause or reply to any part of lectures that they could not catch up on during the course [Pers. Comm. KII–6]. The group discussion suggests that students started to accept online learning when familiar with using platforms and electronic divices. However, some students did not spare time to learn how to use the platform. Both lecturers and students similarly raised the issues of slow internet access. Students who returned to their home town could not regularly or adequately access good service of internet. In addition, many students could not afford to cover the home WIFI connection around 15 dollars per month; they mainly depended on the internet of phone service [Pers. Comm. FDG–1].

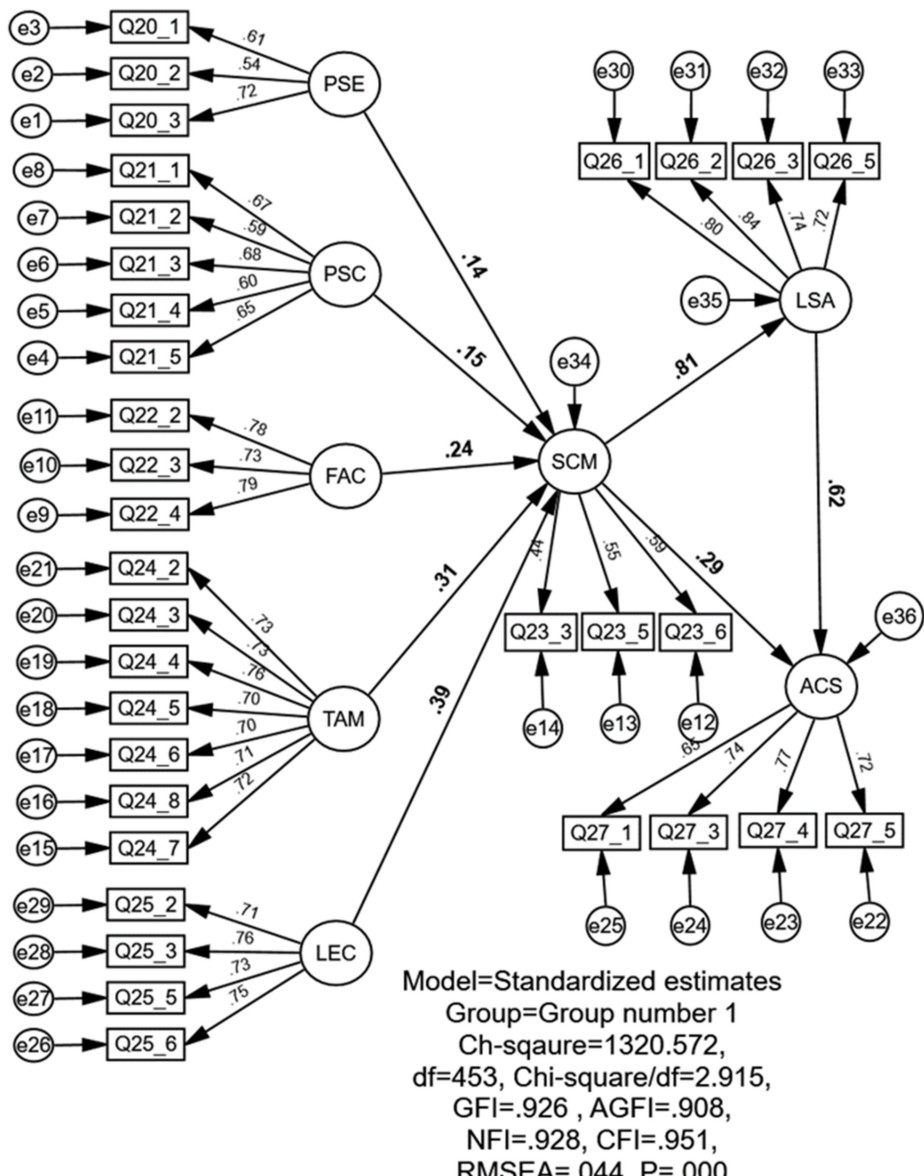

**Figure 5.** The Results of Structural Equation Modeling (SEM). Note: LSA: Learning Satisfaction; PSE: Perceived Self-Efficacy; PSC: Psychological Climate; FAC: Faculty Climate; TAM: Technical Arrangement; LEC: Learning Capability; SCM: Study Commitment; ACS: Academic Support.

Academic support motivated students to pursue online learning; lecturers were responsible for teaching correctly and responding to student needs. It was essential for students and lecturers to have an e-mail account from Microsoft Cambodia because they were using licensed Microsoft Teams for video conferencing. The IT team at RUPP worked with all the faculties to create 39,769 e–mail accounts for all students and lecturers. The IT team provided timely technical assistance for creating institutional e-mail accounts, training among lecturers and students, and preparing instructions on using Microsoft Teams. The University worked closely with lecturers and students to ensure online teaching and learning were smoothly operating [Pers. Comm. KII–4]. If students had high commitment and efforts in adopting a new learning environment, they should not have any problem with online learning. However, few students still felt like they were teaching and learning themselves without any support from the University or lecturers. Meanwhile, some lecturers did not prepare well and translated well into an online format and occasionally dropped students from class without letting them know [Pers. Comm. KII–5].

The model confirms that the students' study commitment played a mediating role. While the study commitment was influenced by learning capability, technical arrangement, psychological climate, faculty climate, and perceived self-efficacy, it was associated with learning satisfaction and academic support. The consultative meeting agrees that students required motivation and support from the University and lecturers. They needed great attention to ensure that students were satisfied with their study performance. During online teaching, students were facing some critical issues, such as limited adaptability capacity, insufficient technical support, computer literacy, and lack of self-motivation [Pers. Comm. CM–1]. Some lecturers and students still had limited knowledge of computer-based training in a virtual classroom. Some lecturers and students remained troubled with computer proficiency in online learning. As a result, lecturers and students lost self-motivation when they spent much time in online classes [Pers. Comm. KII–5]. Technical arrangement and faculty climate established an excellent online learning environment; timely supervision allowed students to follow course schedules and complete tasks and assignments due to deadlines. On the other hand, the study commitment of students at RUPP depended on the technical arrangement, faculty climate, psychological climate, and perceived self-efficacy. The learning capability, perceived self–efficacy, and psychological climate of students were helpful for students to perform novel or challenging tasks during their study online. During the group discussion, students asked for understanding from lecturers if they could not fulfill all the requirements of the lecturers. Some students faced mental health difficulties, stress, and isolation; they struggled to compete for priorities due to the pandemic: work, family, and school work. Some students were bored with being alone and had no interactions with classmates. Students did not have a good study environment or were distracted at home [Pers. Comm. FDG–1]. However, the Department of Psychology delivered consultation services; but the capacity was insufficient. Many students were unwilling to freely discuss their stress and mental issues with the University, and their lecturers. However, they preferred to consult with their classmates, friends, or family members. As a result, the University was not much helpful in assisting students in reducing stress and mental health difficulties among their online learning [Pers. Comm. KII–5].

### 4.3. Factors Influencing Students' Decisions to Have Future Online Learning

The survey reveals that the study performance of the students was significantly reduced from a high degree of satisfaction of in-class learning (WAI = 0.6571) to a moderate degree of satisfaction of online learning (WAI = 0.5541) (*p*-value = 0.000). Overall, 81.4% of the students did not wish online learning in the post-COVID-19 pandemic. The result of Logistic Regress, i.e., a combination of variables, significantly estimated the decision in having online learning in the post–COVID-19 pandemic of $X^2$ = 881.98, d.f. = 1, $N$ = 1002, $p < 0.05$ with four out of the nine proposed variables contributing to all predictions. Table 3 suggests that gender, the effect of online learning, permanent address, and home WIFI connection contributed to the students' decision to have online learning in the post–COVID-19 pandemic.

Overall, 62.5% of the students claim that online learning affected their academic performance. Students with a more negligible effect on their academic performance (29.3%) tended to share a higher proportion of willingness to continue online education than affected students (12.1%). Other factors such as permanent residence in Phnom Penh (22.7%) and the availability of home WIFI connection (24.9%) also contributed to their decision to continue their online education in the post–COVID-19 pandemic. Students with could not access a home WIFI connection (15.4%) and had permanent residences (14.1%) in other provinces were not willing to continue online learning because they were not privileged to this learning approach. As a result, only 18.6% of students wished to continue online learning; male students (22.2%) were more anxious and optimistic towards online education than female students (14.0%). The leading causes of willingness to continue online education include time and money-saving, the current availability of

various practical and flexible platforms for educational purposes, and the creation of an independent learning environment [Pers. Comm. KII–3].

**Table 3.** Factor influencing students to decide in having future online learning.

| Attributes | B | Exp(B) | Standard Error | Wald Statistic | *p*-Value (sig) |
|---|---|---|---|---|---|
| Marital | −0.371 | 0.690 | 0.384 | 0.932 | 0.334 |
| Gender | 0.591 | 1.806 | 0.183 | 10.398 | **0.001** |
| Effect of online learning and study | −1.132 | 0.322 | 0.172 | 43.171 | **0.000** |
| Faculty | −0.295 | 0.745 | 0.185 | 2.539 | 0.111 |
| Home town | −0.112 | 0.894 | 0.237 | 0.224 | 0.636 |
| Permanent address | 0.470 | 1.600 | 0.214 | 4.821 | **0.028** |
| Home WIFI connection | 0.405 | 1.499 | 0.192 | 4.461 | **0.035** |
| Study and work | −0.011 | 0.989 | 0.196 | 0.003 | 0.955 |
| Study time | 0.151 | 1.163 | 0.180 | 0.708 | 0.400 |
| Constant | 0.975 | 2.651 | 0.326 | 8.927 | 0.003 |

Note: d.f. = 1, $N$ = 1002, $p < 0.05$; significant betas are shown in **bold**. −2 Log likelihood or $X^2$ = 881.98; Cox and Snell R Square = 0.075; Nagelkerke R Square = 0.122.

Education at HEIs is in an excellent position to continue the online education because of these factors: (1) maturity of students in taking actions and responding to school requirements remotely, (2) available physical infrastructure and human and financial resources; and (3) more flexible curriculum to adopt web-based teaching [Pers. Comm. KII–2]. At the same time, the MoEYS worked hard to ensure the continuation of administrative work to proceed at all safe teaching and learning procedures. The MoEYS also implemented the SOP, technical guidelines, and instructions to ensure all HEIs operate online teaching and learning in all circumstances [Pers. Comm. KII–3]. The official at the MoEYS suggests that online teaching and learning also helped promote digitalization and the 21st century endorsed by the Ministry. Furthermore, online educational platforms may cost less than teaching operations in the classroom. Online learning can customize learners' environment as it offers opportunities to concentrate and develop a deeper understanding of the content knowledge. The online educational platform was more convenient and flexible. It allows a better balance of work and studies for teachers and the students to set their own pace. It also helps boost learners' career advancement opportunities while taking only the courses. The students can upgrade their knowledge and skills serving their work demands [Pers. Comm. KII–1].

## 5. Discussion

### 5.1. Barriers Reducing Academic Performance during the COVID-19 Pandemic

The survey shows that students' academic performance reduced during the COVID-19 pandemic. Both lecturers and students raised similar issues regarding insufficient electrotonic devices, low access to internet services, and limited knowledge of educational platforms; they used their electronic devices and individual internet services. Lecturers do not have access to sufficient ICT infrastructure, and unstable internet connections occasionally cut the conversation between students and teachers. Lecturers often waste much time repeating their lecturers during poor internet services. Moreover, online teaching and learning are the main barriers to interaction and insufficient practice. Classrooms give better academic performance because lecturers and students can interact and form groups for discussion and sharing. In addition, students in science, technology, and engineering are required to practice or do experiments in laboratories, but the students were not allowed to visit the University during the pandemic. Students in social science, arts, and humanities are unable to do fieldwork or to be involved in practical experiences such as volunteer work and internships. Online learning during the COVID-19 pandemic could only provide lectures, while students also require outside classroom activities. Yet, online teaching and learning were the only alternatives to pursue students' education during the COVID-19 pandemic.

The exiting studies of Lean waste have provided lessons learned and best practices to enhance student engagement and efficient learning environment at Cambodia's HEIs. Khandan and Shannon (2021) investigate Lean wastes within online, hybrid, and face-to-face teaching environments to gain solutions for increasing student engagement. The study at the University of Northampton reveals that online teaching had the greatest sources of waste compared to hybrid and face-to-face education. While significant Lean waste within online and hybrid teaching was unused talents, motion was considered the predominant Lean waste within face-to-face teaching [64]). In Brazil and Portugal, Lean management practice was used to assess its influence on sustainability practice at HEIs. This cross-country perspective recognizes the importance of leadership and systematic version to increase staff engagement. Waste elimination requires a continuous effort through the daily routine, long-term thinking, and student attention [65]. Moreover, educators' new roles have been recommended to reduce Lean waste within online teaching; they have to work with health officers based on intersectional and intersectoral approaches to promote well-being, mitigating the negative impact of the COVID-19 pandemic [66]. In Cambodian HEIs, the hybrid learning environment is the most appropriate solution to reduce Lean waste and improve student engagement. Students cannot take online learning for whole courses; they are also required to be in class for practice and interaction. Support from lecturers and the University would be critical to engage students in teaching and learning and communicate with parents about their academic performance and safety.

The survey confirms that students are delighted with Microsoft Teams (WAI = 0.736) and neutral satisfied with Zoom (WAI = 0.481), Messenger (WAI = 0.481), and Google Meeting (WAI = 0.468). However, lecturers agreed that Zoom has the highest quality and is easier to use; only basic Zoom of 40 minutes is freely available. Lecturers cannot afford to buy a Zoom license. Similarly, the accessible version of Google Meet is also limited to 60 minutes; so some lecturers would not finish their one and a half or three-hour lectures. In general, lecturers used Messenger to communicate with students to assign tasks. The offer and support of Microsoft Cambodia have been timely and helpful for the RUPP to provide online teaching and learning services during the COVID-19 pandemic. Up-to-date, Microsoft Teams has been the most appropriate platform to operate online teaching and learning at RUPP. This platform is designed for educational purposes. Microsoft Teams can be used for conversations, content, assignments, and apps together in one place, letting lecturers establish vibrant learning environments. Moreover, Microsoft Teams can use collaborative classrooms, connect in professional learning communities, and link with students and lecturers, all from a single experience.

At RUPP, licensed Microsoft Teams is being extensively used to assist online teaching and learning; this platform enables lecturers and students to interact more effectively. Since November 2020, the RUPP has promoted the use of Microsoft Teams because education with other platforms such as Zoom and Google Meet require monthly reports and screenshots of teaching-related activities. Lecturers using Microsoft Teams do not need to provide monthly reports because the IT team can manage the teaching and learning process. In November 2021, the RUPP started to code each online class so the IT could better manage the teaching and learning process. Yet, the Vice-Rector realized that the RUPP requires more work and efforts to fully adopt the use of Microsoft Teams as institutionally-managed online learning and teaching processes. To fully institutionalize, the RUPP needs to develop a framework to manage online teaching, learning, and assessment. Moreover, both lecturers and students should be provided with digital infrastructures and digital pedagogy [Pers. Comm. KII–4]. During the fieldwork, lecturers could use the essential functions of Microsoft Teams; they still have limited skills in using the application for interaction, for example, breaking group discussion room. Lecturers and students do not take advantage of this platform because they cannot fully use all those valuable functions. Many lecturers and students are not entirely familiar with or willing to explore the functions of this educational platform.

*5.2. Interaction of Study Commitment of Students during Online Learning*

The result of SEM helps to test the hypothesis of students' study commitment to online education; the discussion is divided into eight sections. First, the survey results argue that learning capability positively contributes to students' study commitment. Zareie and Jafari Navimipour (2016) similarly found that learning capability has significantly influenced employees' commitment [67]. In Pakistan, a study by Rafique, Hameed, and Agha (2018) indicates that learning capability and organizational commitment occurred among the middle managers at the pharmaceutical firms [68]. However, organizational learning capability and organizational commitment have been comprehensively studied in the private sector [69,70]; students' learning capability and study commitment to online learning have not been yet investigated in the educational setting. According to Anzai and Simon (1979), the theory of learning by doing and the theory of the process enable the students to learn in any circumstance. Both theories have engaged in solving problems leading to an increase in their study commitment [71].

Second, this research confirms improved technical arrangements by the universities and academic staff increased a high level of students' study commitment. Lincoln and Kalleberg (1992) propose organizational structures engendering employee commitment [72]. Management and technical support are associated with academic staff's professional development and teaching performance during online learning [73]. In this sense, students' study commitment depends on technical support or arrangement to be highly engaged in their online learning. The organizational support theory proposes that students form a generalized perception of the extent to which the organization values their contributions and cares about their well-being, leading them to achieve a high level of commitment to learning online [74,75].

Third, this research reveals that students' study commitment to achieving a high level of online learning depends on a good environment of the psychological climate. At HEIs, the existing literature is investigating relationships among psychological climate and team commitment [76], affective commitment [77,78], and organizational commitment [79,80]. The psychological climate is an important contextual factor that reflects students' attitudes and behaviors during online learning or physical classes [81]. Moreover, the psychological climate has a substantial impact on student learning attitudes in terms of commitment and morale [80].

Fourth, faculty climate has a significant and positive influence on students' study commitment. According to Robbins and Coulter (2018), in social cognitive theory, individual students are likely to impact the faculty's role in the administration during online learning when they believe in their capability of performing a task or achieving a high level of learning commitment [82]. While Thomas (2008) suggests faculty climate has significantly impacted students' study commitment [83], Uline and Tschannen-Moran (2008) maintain that school climate is one of the most critical factors influencing the students' study commitment [84]. An improved faculty role in establishing a good learning environment and faculty–student interaction motivates students' study commitment. Fifth, this research found that students' study commitment is positively impacted by a high degree of perceived self-efficacy. Perceived self–efficacy has a positive and significant impact on students' study commitment. The self-efficacy theory suggests that individuals with self-efficacy perceptions are motivated for better academic performance and are frequently employed in teachers' commitment research [85].

Sixth, this research concludes the relationship between the student's study commitment and learning satisfaction. It is an essential part of establishing a satisfied learning environment through the integration of social identity theory. Students' study commitment and learning satisfaction are positively associated; an empirical finding of this research is consistent with a study by Alshehri (2017) [86]. At the HEIs, students' study commitment is also a predominant influence in managing students' learning satisfaction [87]. According to Robbins and Coulter (2018), the social identity theory argues that it is a tendency to personally invest in the accomplishments of a group or an individual study performance

and satisfaction [88]. Mingfang and Qi (2018) validate the social identity theory, which implies the effect of online learning and learning satisfaction [89].

Seventh, this research confirms that students' study commitment is an essential factor that intrinsically motivates and encourages academic satisfaction. A finding of this research is in line with a study by Grant-Vallone et al. (2003) [90]. They find a high association between students' study commitment and academic support. Gill et al. (2008) discovered a positive relationship between academic support and perceived student learning satisfaction [88]. Eighth, this research extends academic support and emotional bonding theories to elucidate the perceived student satisfaction of online learning at Cambodia's HEIs. Academic support has a positive and significant impact on learning satisfaction. The existing literature review [91–93] found that academic support has enhanced a high level of students' learning satisfaction. Overall, the results of SEM reveal that conceptual model to explore key consequences and antecedents of students' study commitment to online learning are explained by various theoretical backgrounds, including social learning theory, self–efficacy theory, social identity theory, and social cognitive theory.

*5.3. The Implication of Online Teaching and Teaching in the Post–COVID-19 Pandemic*

In 2014, MoEYS formulated the Policy on Higher Education Vision 2030. This policy aims to ensure that students can obtain knowledge, skills, and competency in response to socio-economic needs and the labor market. Therefore, adopting online teaching and learning at HEIs is the only choice in achieving this national policy. Students have to access and pursue their education in any circumstance; online teaching and learning have gradually emerged during the COVID-19 pandemic. At Cambodia's HEIs, online education can be effectively operated when the MoEYS and the universities work together to develop a national framework, guideline, and procedure to ensure lecturers' competency and students' familiarity in online education. The legal framework should be designed to establish an effective online learning environment by (1) ensuring equity and inclusive access in education to all lecturers and students, (2) ensuring the curriculum is responsive and adaptive in a flexible approach; and (3) building capacity of lecturers and raise awareness of students to be familiar with online teaching and learning.

The adaption of online teaching and learning at Cambodia's HEIs should start with policy support by the MoEYS to develop all required frameworks, guidelines, and procedures. At the same time, the HEIs should allocate their annual budgets to increase the availability of technical and financial resources for establishing a system to operate online teaching and learning. Today, the COVID-19 pandemic has disrupted education for more than two years; new emerging challenges may also occur in the future. Therefore, the adaptation of online teaching and learning must be the only choice for continuing students' education during the destruction. According to the Vice-Rector in charge of students and ICT, the RUPP is moving forward to continue online teaching and learning. Shortly, the RUPP should revisit its curriculum and integrate online education between 10% and 20% of the total subjects required in each field. The RUPP should also consider more courses in classrooms for first- and fourth-year students. In contrast, more online courses must be allocated for sophomore and junior students. By doing so, first-year students are more familiar with the learning environment, and seniors can practice and do experiments before the final examination.

The adaption of Microsoft Teams by the RUPP as a platform for the institutionally-managed online teaching and learning process is a cost–effective approach. While Microsoft Cambodia has generously provided free and unlimited licensed usage, the RUPP should prioritize its investment in building the capacity of lecturers and students and supporting services to adopt online teaching and learning professionally and effectively. In particular, the RUPP should work to: (1) improve ICT infrastructure and (2) develop the learning management system (LMS); and formulate a regulation on E-Learning procedure and contents. Moreover, a guideline is required to develop for monitoring and evaluating the new regular, building staff's capacity to use the platforms, and for online class management.

In the future, a new normal teaching and learning environment can not be in the classroom fully; online must be added. A 10 million US dollar project funded by Swedish International Development Cooperation Agency's (SIDA) is implementing a pilot phase to improve the RUPP's ICT infrastructure and assist scientific research and postgraduate training at the Faculty of Science and Engineering. By the end of 2023, the RUPP would have a robust ICT infrastructure to institutionalize the online learning and teaching process fully. However, the RUPP still requires equipment with a video conference in each classroom. The assessment of video conference demands and costs is underway, so the RUPP mobilizes investment from domestic funds and international sources. Up–to–date, the RUPP has not decided if students should be in the classroom or online classes for the coming enrollment in March 2021. A possible scenario should be more classroom-based teaching for first– and fourth–year students. Sophomores and junior students should be more online. To be in line with the MoEYS's SOP, a hybrid teaching and learning option should also be adopted: half (20 students) in the classroom and the other half (20 students) taking online sessions. They should take turns to be at the school, but for practical work and experiment assessment there should be physical access to the University.

## 6. Conclusions

Based on our findings at the RUPP, along with some additional insights about the negative impacts of online learning and teaching at the HEIs in Cambodia, we conclude the following: (1) online teaching and learning during the COVID-19 pandemic at the RUPP has been adopted through two stages: individually-managed and institutionally-managed online teaching and learning process. First, lecturers used available and experienced platforms to resume the class. Second, Microsoft Teams was adopted by the RUPP as an individually-managed online teaching and learning process after long-period educational disruption due to the COVID-19 outbreak. As a result, most of the students at the RUPP (99.2%) experienced Microsoft Teams, followed by Zoom, Google Meet, Messenger, and Skype. The smartphone (86.6%) was the more popular electronic device for online learning with the reason of inaccessibility to home WIFI connection. While students without laptops could not practice for the fields that require computer programs, students without smartphones could not access lectures. (2) Students' study commitment is well–predicted by the SEM model as a mediating variable to associate between independent variables (i.e., learning capability, technical arrangement, psychological climate, faculty climate, and perceived self–efficacy) and dependent ones (i.e., learning satisfaction and academic support). The students' study commitment played a vital role in improving their learning satisfaction. Meanwhile, academic support is considered as one of the most influencing factors increasing student learning during their online learning. (3) Overall, 81.4% of the students did not wish to pursue online learning in the post–COVID-19 pandemic because 62.5% of them revealed that their academic performance was affected during online learning. As a result, the study performance of the students significantly reduced from a high degree of satisfaction with in-class learning to a moderate degree of satisfaction with online learning. The Logistic regression predicts that gender, the effect of online learning, permanent address, and home WIFI connection contributed to the students' willingness to have online education in the post–COVID-19 pandemic. The paper addresses a gap in the literature by contributing students' study commitment at the Cambodia's HEIs about learning capability, technical arrangement, psychological climates, faculty climates, perceived self-efficacy, academic support, and learning satisfaction [27–33]. The results of SEM argue that a conceptual model to explore key consequences and antecedents of students' study commitment to online learning is explained by various theoretical backgrounds, including social learning theory, self-efficacy theory, social identity theory, and social cognitive theory. The understanding of students' study commitment at the HEIs is also applicable for developing countries because their teaching and learning share similar situations, constraints, and lecturer and student backgrounds.

**Author Contributions:** Conceptualization, C.C. and S.S.; methodology, S.S., C.C. and V.S.; software, V.S., C.C. and S.S.; validation, C.C. and S.S.; formal analysis, V.S., S.S. and C.C.; investigation, C.C. and S.S.; resources, C.C.; data curation, S.S. and C.C.; writing–original draft preparation, C.C. and S.S.; writing–review and editing, C.C. and S.S.; visualization, C.C. and S.S.; supervision, C.C. and S.S.; project administration, C.C. and S.S.; funding acquisition, C.C. All authors have read and agreed to the published version of the manuscript.

**Funding:** Royal University of Phnom Penh, Russian Federation Boulevard, Khan Toul Kork, Phnom Penh 12150, Cambodia.

**Institutional Review Board Statement:** Not applicable.

**Informed Consent Statement:** Not applicable.

**Data Availability Statement:** Data is based on a survey among students collected at the RUPP.

**Acknowledgments:** The author greatly appreciates the valuable comments and suggestions in preparing this paper for publication from the editorial board of Sustainability, the three anonymous reviewers. Thanks to Tak Kean (Vice–Rector in charge of students and ICT, RUPP), MoEYS officers (Department of Planning, Department of Policy, and Department of Higher Education, MoYES), students and lecturers at RUPP for providing information and supporting during field work.

**Conflicts of Interest:** The authors declare no conflict of interest.

## Appendix A  Extra Tables and Figures

**Table A1.** Questions used in each research contract and scale. Adopted from Froman et al., 2020 [94].

| Code | Items | Research Construct | Scale |
|---|---|---|---|
| Q25_1 | I can complete course assignments in a timely manner. | Learning capability | 1 = Slightest constraints<br>2 = Less constraints<br>3 = Moderate constraints<br>4 = High constraints<br>5 = Very high constraints |
| Q25_2 | I have enough ability to complete coursework. | | |
| Q25_3 | I am not capable to fully focus or pay attention to remote instruction or activities. | | |
| Q25_4 | I can perform better if I am attending face-to-face learning. | | |
| Q25_5 | I do not understand the course because I do not have clear expectations around course/assignment requirements. | | |
| Q25_6 | I am finding time to participate in all the classes (e.g., live-streaming lectures or videoconferencing at a set time) to follow the course well. | | |
| Q25_7 | I have difficulty to understand course lessons or activities because they have not translated well to a remote environment. | | |
| Q25_8 | I am completing class meetings and schedules. | | |
| Q25_9 | I have a personal preference for face-to-face learning. | | |
| Q24_1 | I am keeping up with course work through different online platforms. | Technical arrangement | 1 = Slightest constraints<br>2 = Less constraints<br>3 = Moderate constraints<br>4 = High constraints<br>5 = Very high constraints |
| Q24_2 | Internet connection is affecting my grades/performance well in class. | | |
| Q24_3 | I am sparing my time to fix issues incurred during online learning. | | |
| Q24_4 | I am communicating with lecturers through online applications. | | |
| Q24_5 | I am communicating with classmates through online applications. | | |
| Q24_6 | Insufficient technology possibly delays in graduating/completing my program. | | |

**Table A1.** *Cont.*

| Code | Items | Research Construct | Scale |
|---|---|---|---|
| Q24_7 | The online platform does not include on-campus activities. | | |
| Q24_8 | Online application is affecting grading structure (e.g., pass/fail, credit/no credit). | | |
| Q24_9 | I have a concern regarding online privacy, protection of personal data. | | |
| Q24_10 | I have to deal with security/privacy in taking online exams. | | |
| Q24_11 | My accommodations are not accessible to an internet connection. | | |
| Q24_12 | I cannot carry out the experiment, internship, or practicum requirements through online learning. | | |
| Q21_1 | I am facing mental health difficulties, stress, and isolation. | | |
| Q21_2 | I have competing priorities due to pandemic, work, family, and school work. | | 1 = Slightest constraints |
| Q21_3 | I do not have an adequate studying environment or distractions at home. | **Psychological climate** | 2 = Less constraints<br>3 = Moderate constraints |
| Q21_4 | I am facing job security or financial concerns, or housing concerns. | | 4 = High constraints<br>5 = Very high constraints |
| Q21_5 | I am bored with being alone and miss interactions with classmates. | | |
| Q22_1 | I find it difficult to respond to lecturers' demands or heavy coursework. | | 1 = Slightest constraints |
| Q22_2 | I do not receive sufficient lecturers' support. | **Faculty climates** | 2 = Less constraints<br>3 = Moderate constraints |
| Q22_3 | I prefer face-to-face communication with lecturers. | | 4 = High constraints |
| Q22_4 | I find that lecturers need training regarding online teaching. | | 5 = Very high constraints |
| Q20_1 | I find it difficult to teach myself or learn remotely. | | |
| Q20_2 | I prefer hands-on learning or access to the lab and course tools. | **Perceived self–efficacy** | 1 = Strongly Disagree<br>5 = Strong Agree |
| Q20_3 | I find it difficult to access student support services/technology. | | |
| Q23_1 | My graduation is delayed due to canceled classes, retakes, and transfer concerns. | | |
| Q23_2 | I have grade concerns, unclear grading, grades dropping after the move to remote learning. | | 1 = Slightest constraints |
| Q23_3 | I am patient enough to deal with technology issues during online study. | **Study Commitment** | 2 = Less constraints<br>3 = Moderate constraints |
| Q23_4 | I keep connecting to online platforms; however, the internet is slow. | | 4 = High constraints<br>5 = Very high constraints |
| Q23_5 | I am adopting my home as a new classroom. | | |
| Q23_6 | I am trying to understand how to take online courses effectively. | | |
| Q26_1 | I do not feel like I am receiving the same quality of education. | | 1 = Slightest constraints |
| Q26_2 | Remote instruction is not operating well. | **Learning satisfaction** | 2 = Less constraints<br>3 = Moderate constraints |
| Q26_3 | Students are facing unclear instruction on assignments, exams, and/or quizzes. | | 4 = High constraints<br>5 = Very high constraints |
| Q26_4 | Students' workloads are increasing. | | |

**Table A1.** *Cont.*

| Code | Items | Research Construct | Scale |
|---|---|---|---|
| Q26_5 | Online learning creates tutoring in-person or disconnected from lecturers. | | |
| Q26_6 | Students are feeling ignored. | | |
| Q27_1 | University/lecturers are not being responsive to questions or student needs. | | |
| Q27_2 | I am generally getting support from University/lecturers to move online. | | |
| Q27_3 | Lessons provided by University/lecturers are not happening, inadequate for learning, or should not be required to attend. | | 1 = Slightest constraints |
| Q27_4 | Students feel like they are teaching and learning themselves without any support from University/lecturers. | Academic support | 2 = Less constraints<br>3 = Moderate constraints<br>4 = High constraints |
| Q27_5 | University/lecturers are struggling with Canvas or transition to online. | | 5 = Very high constraints |
| Q27_6 | Program or course work did prepare well because the University/lecturers did not translate well into an online format. | | |
| Q27_7 | University/lecturers are not involved in teaching properly. | | |
| Q27_8 | University/lecturers dropped students from class without letting them know. | | |

**Table A2.** The Results of Structural Equation Modeling (SEM).

| Constructs | Indicators | | Standardized Coefficient ($\beta$) | *t*-Value | *p*-Value |
|---|---|---|---|---|---|
| **Perceived Self–Efficacy** | $\rightarrow$ | Q20_3 | 0.717 | A | *** |
| | $\rightarrow$ | Q20_2 | 0.543 | 11.863 | *** |
| | $\rightarrow$ | Q20_1 | 0.607 | 12.28 | *** |
| **Psychological Climate** | $\rightarrow$ | Q21_5 | 0.654 | A | *** |
| | $\rightarrow$ | Q21_4 | 0.600 | 15.519 | *** |
| | $\rightarrow$ | Q21_3 | 0.676 | 17.552 | *** |
| | $\rightarrow$ | Q21_2 | 0.594 | 15.758 | *** |
| | $\rightarrow$ | Q21_1 | 0.675 | 17.453 | *** |
| **Faculty Climate** | $\rightarrow$ | Q22_4 | 0.790 | A | *** |
| | $\rightarrow$ | Q22_3 | 0.729 | 19.132 | *** |
| | $\rightarrow$ | Q22_2 | 0.776 | 20.124 | *** |
| **Study Commitment** | $\rightarrow$ | Q23_6 | 0.594 | A | *** |
| | $\rightarrow$ | Q23_5 | 0.553 | 18.826 | *** |
| | $\rightarrow$ | Q23_3 | 0.443 | 17.152 | *** |

**Table A2.** *Cont.*

| Constructs | | Indicators | Standardized Coefficient (β) | *t*-Value | *p*-Value |
|---|---|---|---|---|---|
| **Technical Arrangement** | → | Q24_7 | 0.716 | A | *** |
| | → | Q24_8 | 0.715 | 21.771 | *** |
| | → | Q24_6 | 0.698 | 21.234 | *** |
| | → | Q24_5 | 0.699 | 21.372 | *** |
| | → | Q24_4 | 0.757 | 23.056 | *** |
| | → | Q24_3 | 0.730 | 22.133 | *** |
| | → | Q24_2 | 0.730 | 22.147 | *** |
| **Academic Support** | → | Q27_5 | 0.722 | A | *** |
| | → | Q27_4 | 0.773 | 22.286 | *** |
| | → | Q27_3 | 0.738 | 21.368 | *** |
| | → | Q27_1 | 0.648 | 21.149 | *** |
| **Learning Capability** | → | Q25_6 | 0.748 | A | *** |
| | → | Q25_5 | 0.726 | 21.853 | *** |
| | → | Q25_3 | 0.762 | 23.474 | *** |
| | → | Q25_2 | 0.713 | 21.43 | *** |
| **Learning Satisfaction** | → | Q26_2 | 0.839 | 28.746 | *** |
| | → | Q26_3 | 0.741 | 24.723 | *** |
| | → | Q26_5 | 0.720 | 21.54 | *** |
| | → | Q26_1 | 0.799 | A | *** |
| **Path Relationships** | | | | | |
| H1: Learning Capability → Study Commitment | | | 0.39 | 6.33 | *** |
| H2: Technical Arrangement → Study Commitment | | | 0.31 | 5.126 | *** |
| H3: Psychological Climate → Study Commitment | | | 0.15 | 3.093 | 0.002 |
| H4: Faculty Climate → Study Commitment | | | 0.24 | 5.797 | *** |
| H5: Perceived Self-Efficacy → Study Commitment | | | 0.14 | 3.990 | *** |
| H6: Study Commitment → Learning Satisfaction | | | 0.81 | 17.33 | *** |
| H7: Study Commitment → Academic Support | | | 0.29 | 5.296 | *** |
| H8: Academic Support → Learning Satisfaction | | | 0.62 | 10.54 | *** |
| **Goodness–of–Fit Index** | | | | | |
| $\chi^2$/D.F = 2.915 | | | | | |
| GFI = 0.926 | | | | | |
| AGFI = 0.908 | | | | | |
| NFI = 0.928 | | | | | |
| CFI = 0.951 | | | | | |
| RMSEA = 0.044 | | | | | |

Note: A = parameter regression weight was fixed at 1.000 and significant level of *p*-value < 0.05 and *t*-value > 1.96. *** *p* < 0.001.

**Table A3.** The Results of Common Method Variance (CMV).

| Component | Initial Eigenvalues | | | Extraction Sums of Squared Loadings | | |
|---|---|---|---|---|---|---|
| | Total | % of Variance | Cumulative % | Total | % of Variance | Cumulative % |
| 1 | 20.426 | 36.475 | 36.475 | 20.426 | 36.475 | 36.475 |
| 2 | 2.826 | 5.046 | 41.521 | | | |
| 3 | 2.016 | 3.600 | 45.121 | | | |
| 4 | 1.894 | 3.383 | 48.504 | | | |
| 5 | 1.651 | 2.948 | 51.452 | | | |
| 6 | 1.483 | 2.648 | 54.100 | | | |
| 7 | 1.319 | 2.356 | 56.456 | | | |
| 8 | 1.275 | 2.276 | 58.732 | | | |
| 9 | 1.143 | 2.042 | 60.774 | | | |
| 10 | 1.083 | 1.934 | 62.708 | | | |
| 11 | 1.039 | 1.856 | 64.564 | | | |
| 12 | 0.851 | 1.520 | 66.084 | | | |
| 13 | 0.805 | 1.437 | 67.521 | | | |
| 14 | 0.786 | 1.404 | 68.925 | | | |
| 15 | 0.757 | 1.351 | 70.276 | | | |
| 16 | 0.717 | 1.281 | 71.557 | | | |
| 17 | 0.676 | 1.207 | 72.764 | | | |
| 18 | 0.641 | 1.144 | 73.908 | | | |
| 19 | 0.626 | 1.118 | 75.026 | | | |
| 20 | .605 | 1.081 | 76.106 | | | |
| 21 | 0.580 | 1.036 | 77.143 | | | |
| 22 | 0.561 | 1.001 | 78.144 | | | |
| 23 | 0.549 | 0.981 | 79.125 | | | |
| 24 | 0.539 | 0.963 | 80.088 | | | |
| 25 | 0.521 | 0.931 | 81.019 | | | |
| 26 | 0.507 | 0.906 | 81.924 | | | |
| 27 | 0.500 | 0.892 | 82.816 | | | |
| 28 | 0.479 | 0.855 | 83.672 | | | |
| 29 | 0.460 | 0.822 | 84.494 | | | |
| 30 | 0.458 | 0.819 | 85.312 | | | |
| 31 | 0.437 | 0.781 | 86.093 | | | |
| 32 | 0.426 | 0.761 | 86.854 | | | |
| 33 | 0.422 | 0.754 | 87.608 | | | |
| 34 | 0.405 | 0.724 | 88.331 | | | |
| 35 | 0.399 | 0.712 | 89.043 | | | |
| 36 | 0.380 | 0.679 | 89.722 | | | |
| 37 | 0.376 | 0.671 | 90.393 | | | |
| 38 | 0.366 | 0.653 | 91.046 | | | |
| 39 | 0.361 | 0.645 | 91.691 | | | |

**Table A3.** *Cont.*

| Component | Initial Eigenvalues | | | Extraction Sums of Squared Loadings | | |
| :---: | :---: | :---: | :---: | :---: | :---: | :---: |
| | **Total** | **% of Variance** | **Cumulative %** | **Total** | **% of Variance** | **Cumulative %** |
| 40 | 0.353 | 0.631 | 92.322 | | | |
| 41 | 0.344 | 0.614 | 92.936 | | | |
| 42 | 0.340 | 0.608 | 93.544 | | | |
| 43 | 0.335 | 0.597 | 94.141 | | | |
| 44 | 0.315 | 0.562 | 94.703 | | | |
| 45 | 0.300 | 0.536 | 95.239 | | | |
| 46 | 0.299 | 0.533 | 95.772 | | | |
| 47 | 0.294 | 0.525 | 96.297 | | | |
| 48 | 0.280 | 0.500 | 96.797 | | | |
| 49 | 0.266 | 0.476 | 97.273 | | | |
| 50 | 0.258 | 0.461 | 97.734 | | | |
| 51 | 0.248 | 0.443 | 98.177 | | | |
| 52 | 0.240 | 0.429 | 98.606 | | | |
| 53 | 0.222 | 0.396 | 99.002 | | | |
| 54 | 0.209 | 00.372 | 99.374 | | | |
| 55 | 0.180 | .321 | 99.695 | | | |
| 56 | 0.171 | 0.305 | 100.000 | | | |

Extraction Method: Principal Component Analysis.

**Table A4.** The Results of Mediating Effects by Sobel's Test.

| Mediating Effects | Standardized Coefficient * | Sobel's Test | | | | | | Results |
| :---: | :---: | :---: | :---: | :---: | :---: | :---: | :---: | :---: |
| | | **a** | **$S_a$** | **b** | **$S_b$** | ***z*-Test ** ** | ***p*-Value** | |
| LEC → SCM → LSA | 0.052 | 0.114 | 0.060 | 0.458 | 0.070 | 1.8246 | 0.0680 | Disconfirmed |
| LEC → SCM → ACS | 0.112 | 0.388 | 0.048 | 0.288 | 0.065 | 3.8854 | 0.0001 | Confirmed |
| TAM → SCM → LSA | 0.076 | 0.142 | 0.071 | 0.534 | 0.102 | 1.8683 | 0.0610 | Disconfirmed |
| TAM → SCM → ACS | 0.035 | 0.278 | 0.053 | 0.127 | 0.104 | 1.1893 | 0.2343 | Disconfirmed |
| FAC → SCM → LSA | 0.274 | 0.302 | 0.045 | 0.907 | 0.118 | 5.0554 | 0.0000 | Confirmed |
| FAC → SCM → ACS | 0.057 | 0.226 | 0.033 | 0.250 | 0.080 | 2.8430 | 0.0044 | Confirmed |
| PSC → SCM → LSA | 0.729 | 0.496 | 0.061 | 1.470 | 0.245 | 4.8279 | 0.0000 | Confirmed |
| PSC → SCM → ACS | 0.080 | 0.191 | 0.045 | 0.419 | 0.090 | 3.1366 | 0.0017 | Confirmed |
| PSE → SCM → LSA | 0.198 | 0.226 | 0.041 | 0.874 | 0.070 | 5.0426 | 0.0000 | Confirmed |
| PSE → SCM → ACS | 0.133 | 0.164 | 0.035 | 0.812 | 0.065 | 4.3872 | 0.0000 | Confirmed |

Note: * Standardized Coefficient for mediating effect = a × b; ** z-test = $\frac{ab}{\sqrt{b^2 SE_a^2 + a^2 SE_b^2}}$.

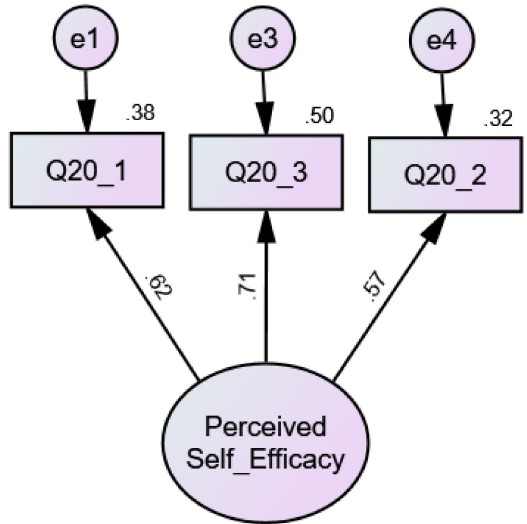

**Figure A1.** First–Order Factor Model–The Result of Perceived Self–Efficacy.

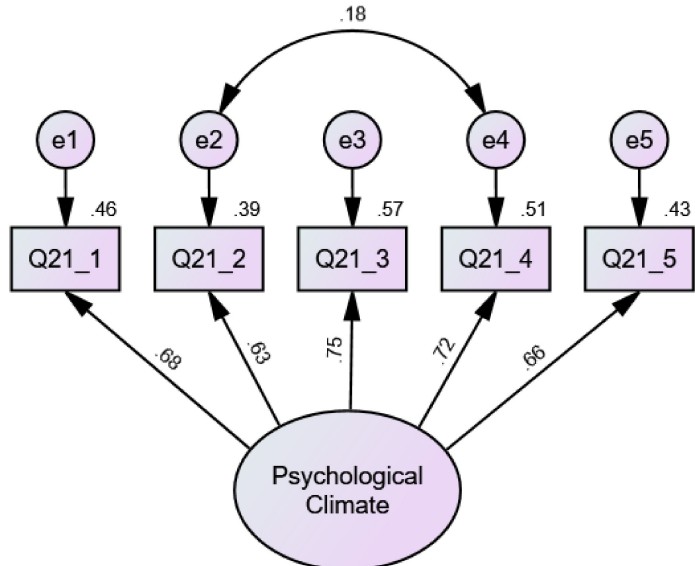

**Figure A2.** First–Order Factor Model–The Result of Psychological Climate.

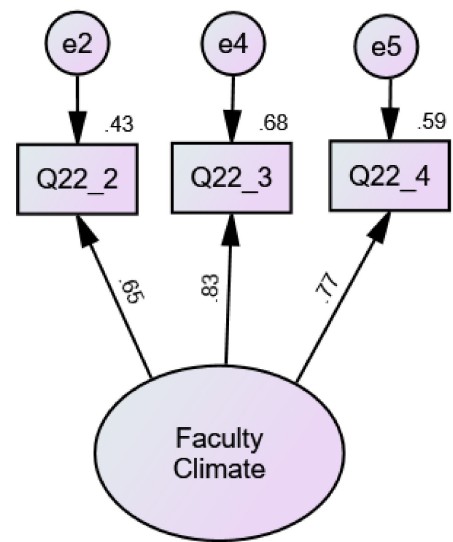

Model=Standardized estimates
Group=Group number 1
Ch-sqaure=.000,
df=0, Chi-square/df=\cmindf,
GFI=1.000 , AGFI=\AGFI,
NFI=\NFI, CFI=\CFI,
RMSEA=\RMSEA, P=\P

**Figure A3.** First–Order Factor Model–The Result of Faculty Climate.

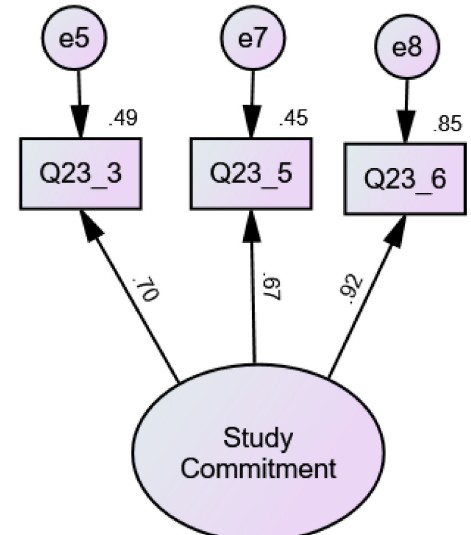

Model=Standardized estimates
Group=Group number 1
Ch-sqaure=.000,
df=0, Chi-square/df=\cmindf,
GFI=1.000 , AGFI=\AGFI,
NFI=\NFI, CFI=\CFI,
RMSEA=\RMSEA, P=\P

**Figure A4.** First–Order Factor Model–The Result of Study Commitment.

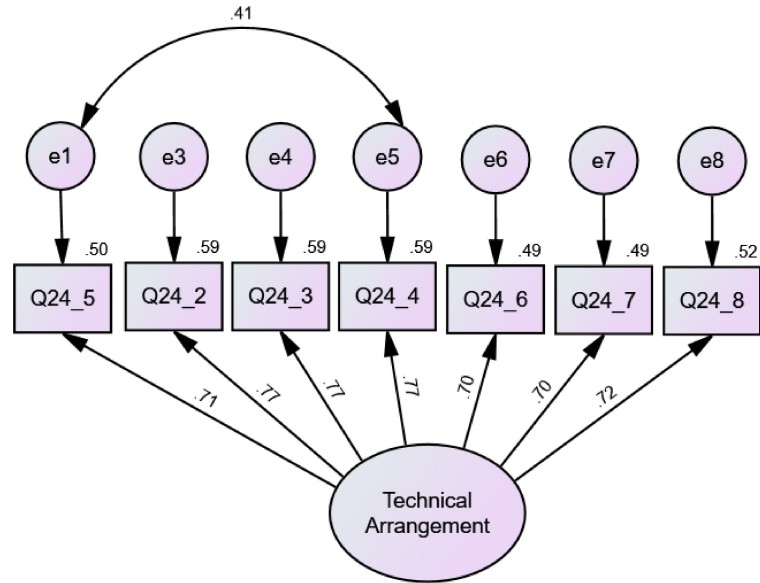

Figure A5. First–Order Factor Model–The Result of Technical Arrangement.

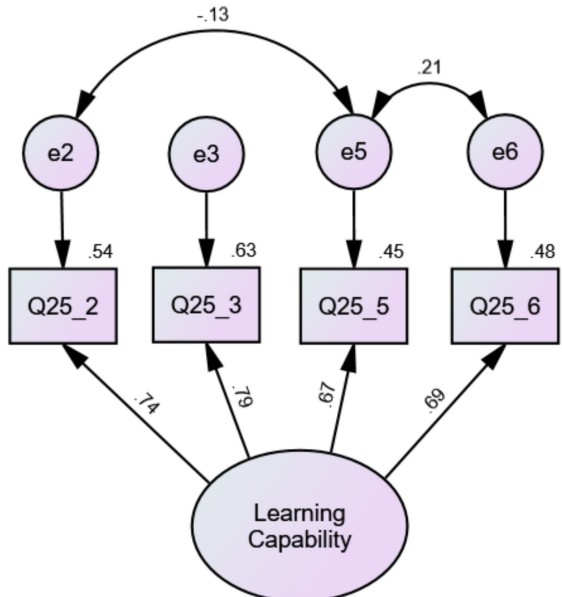

Figure A6. First–Order Factor Model–The Result of Learning Capability.

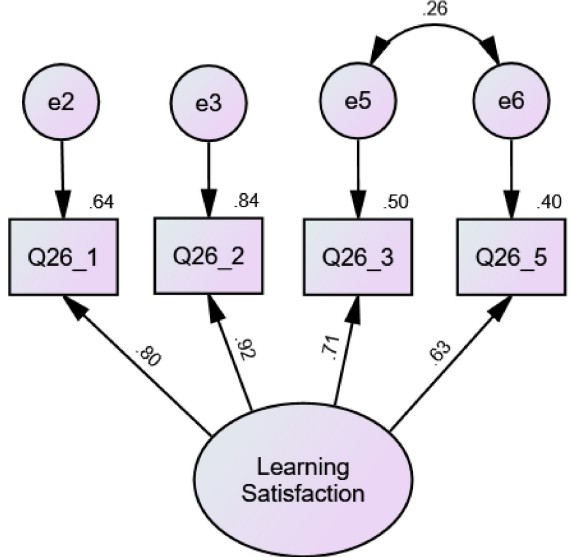

Figure A7. First–Order Factor Model–The Result of Learning Satisfaction.

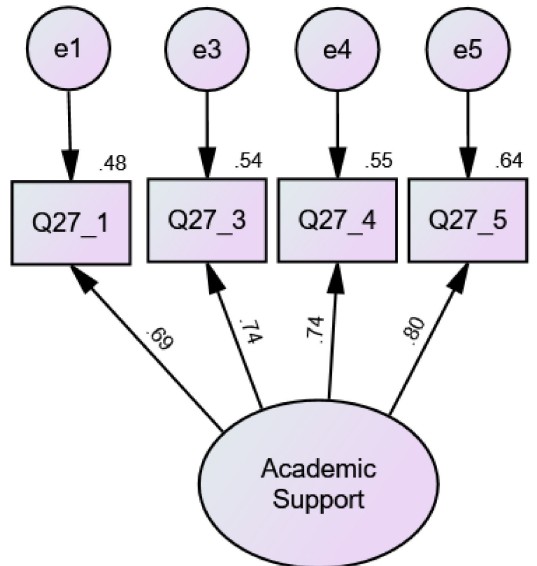

Figure A8. First–Order Factor Model–The Result of Academic Support.

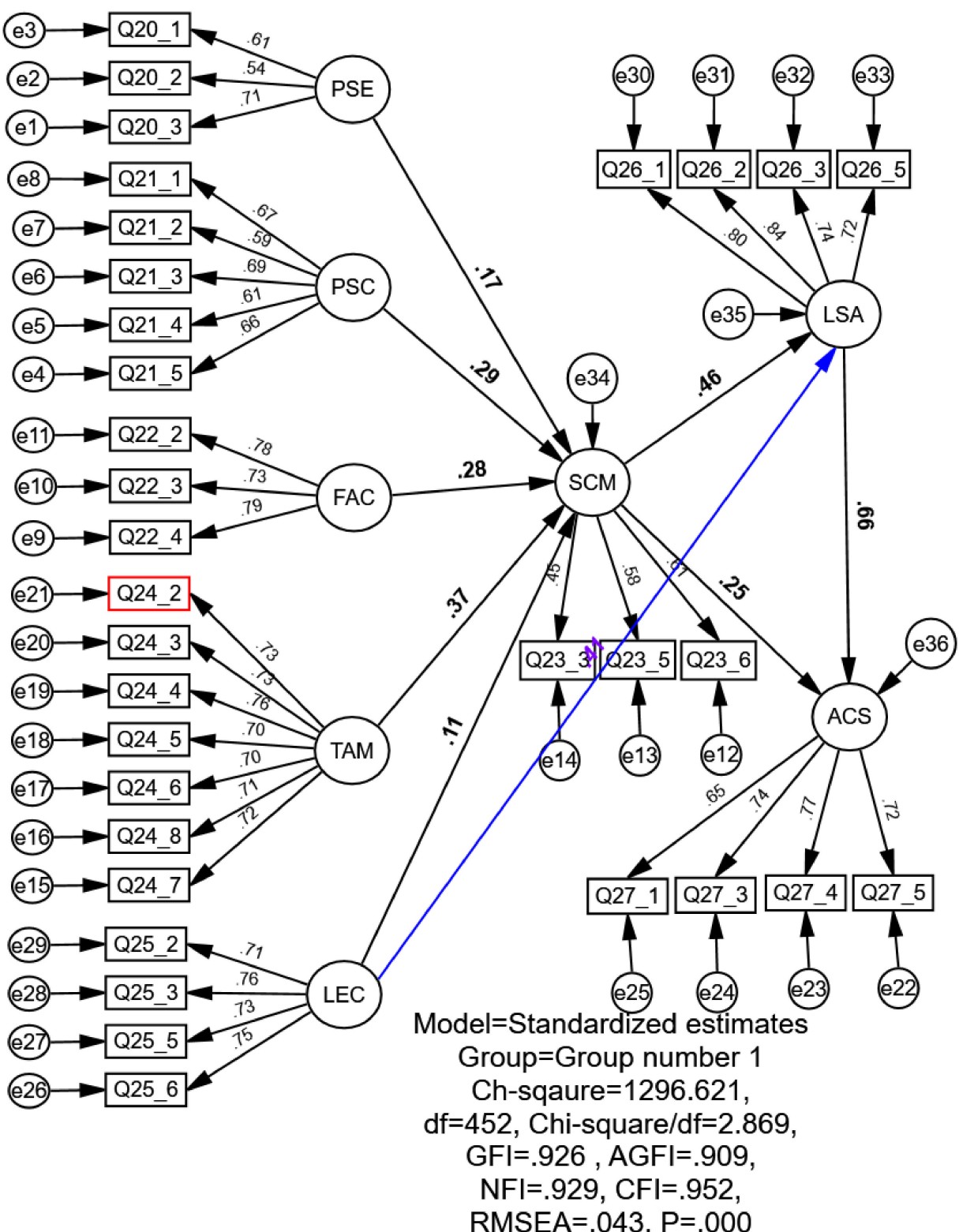

**Figure A9.** The revised model of mediating test: LEC → SCM → LSA (the blue line color is suggested relation for mediating effect).

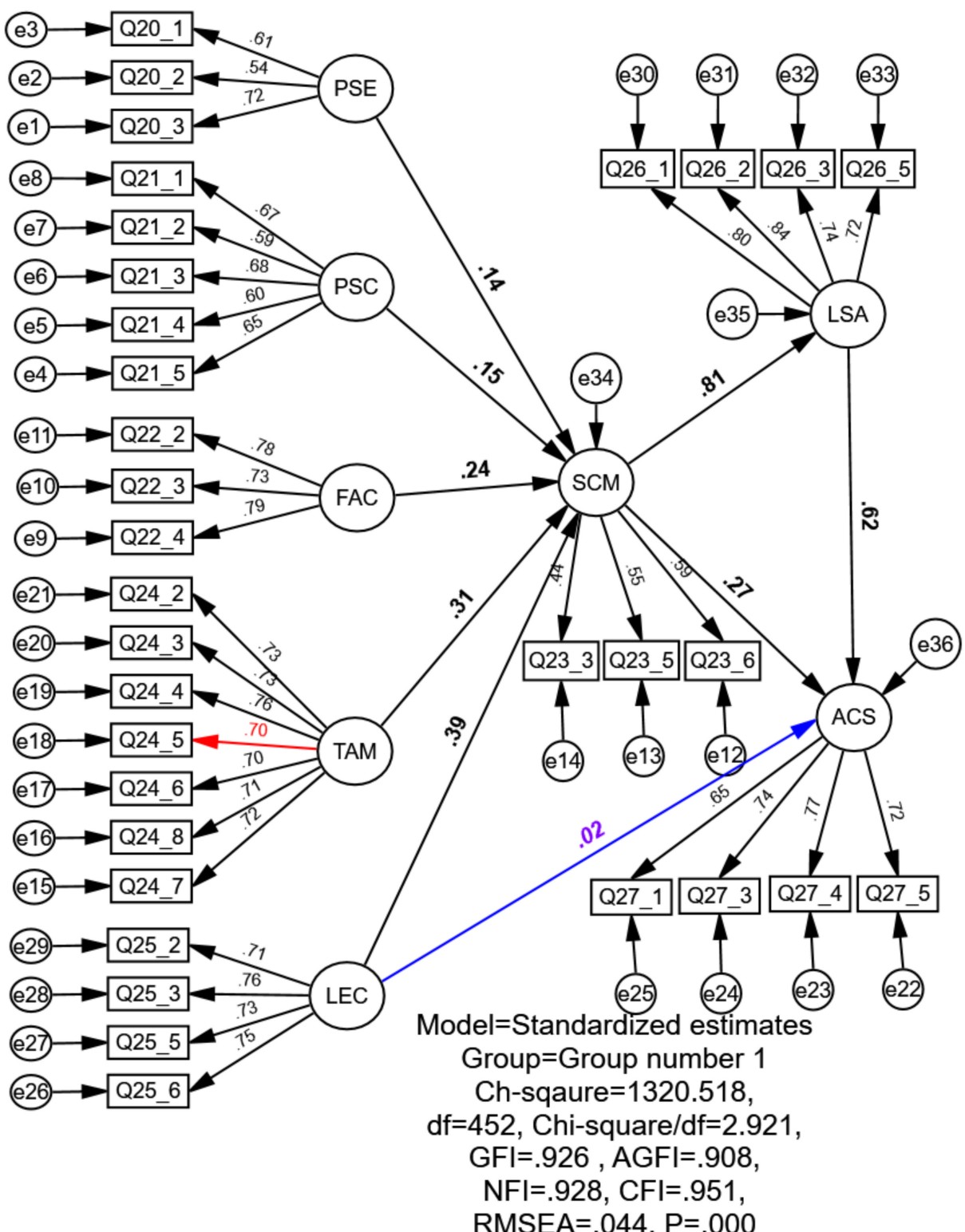

**Figure A10.** The revised model of mediating test: LEC → SCM → ACS (the blue line color is suggested relation for mediating effect).

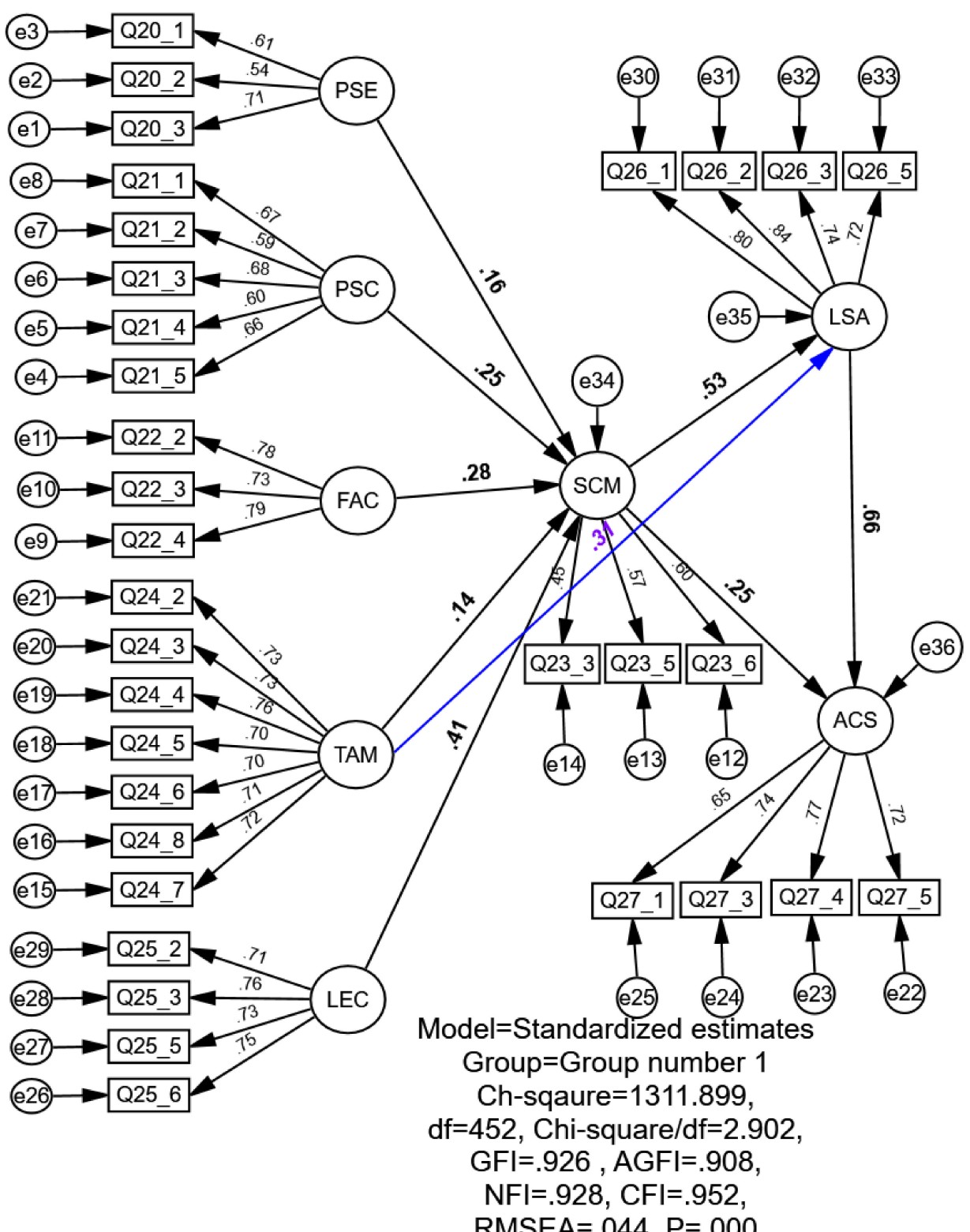

**Figure A11.** The revised model of mediating test: TAM → SCM → LSA (the blue line color is suggested relation for mediating effect).

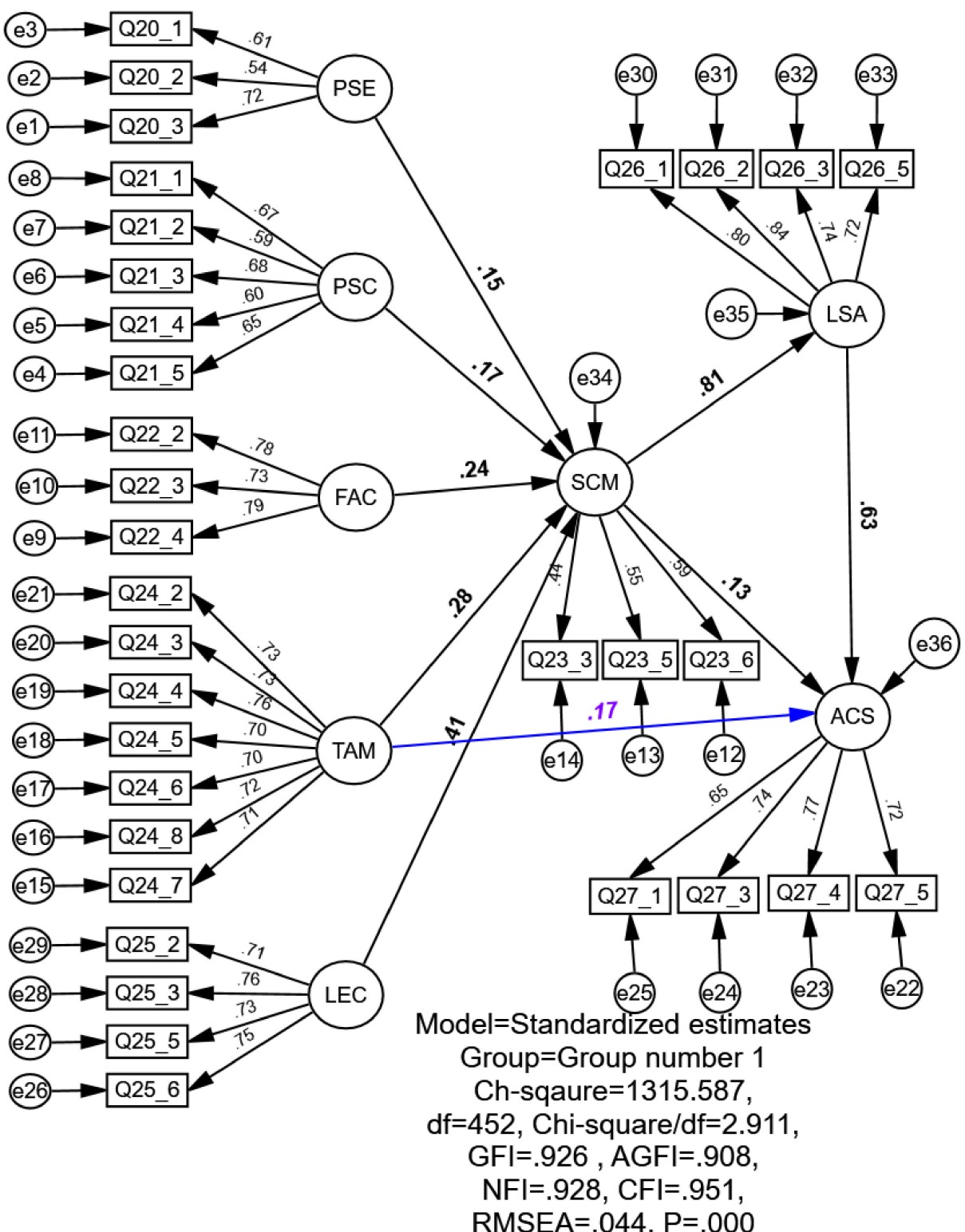

**Figure A12.** The revised model of mediating test: TAM → SCM → ASC (the blue line color is suggested relation for mediating effect).

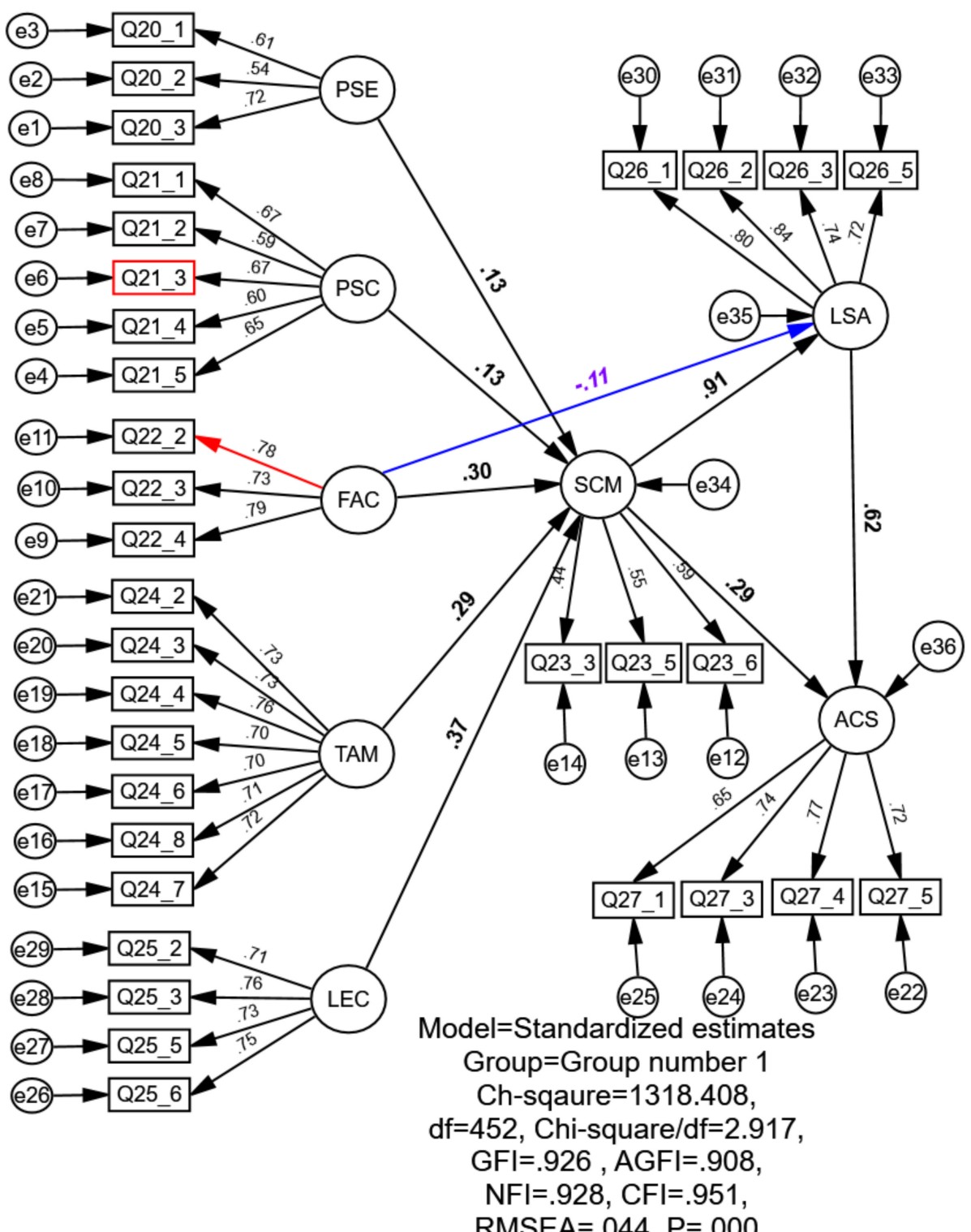

**Figure A13.** The revised model of mediating test: FAC → SCM → LSA (the blue line color is suggested relation for mediating effect).

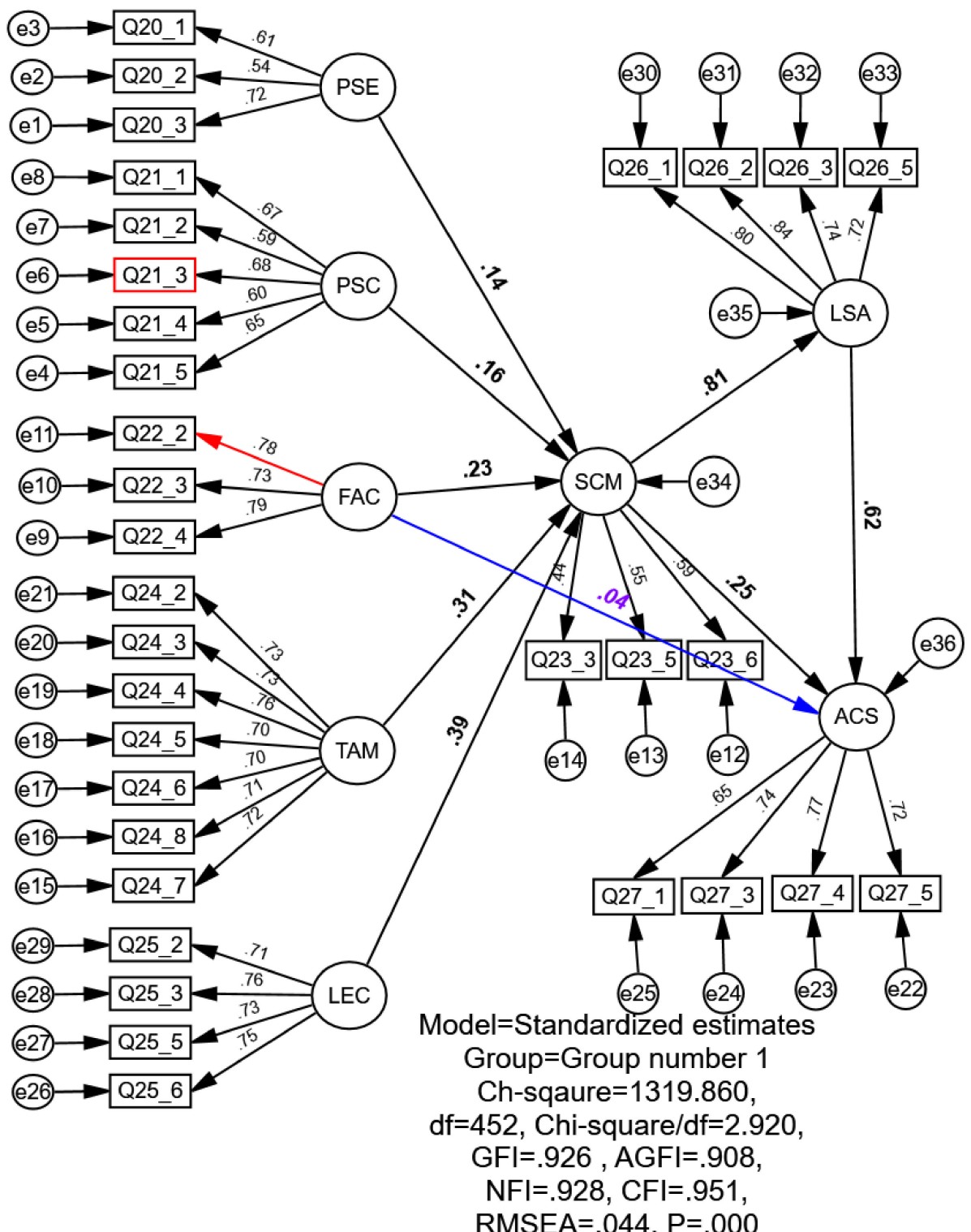

**Figure A14.** The revised model of mediating test: FAC → SCM → ACS (the blue line color is suggested relation for mediating effect).

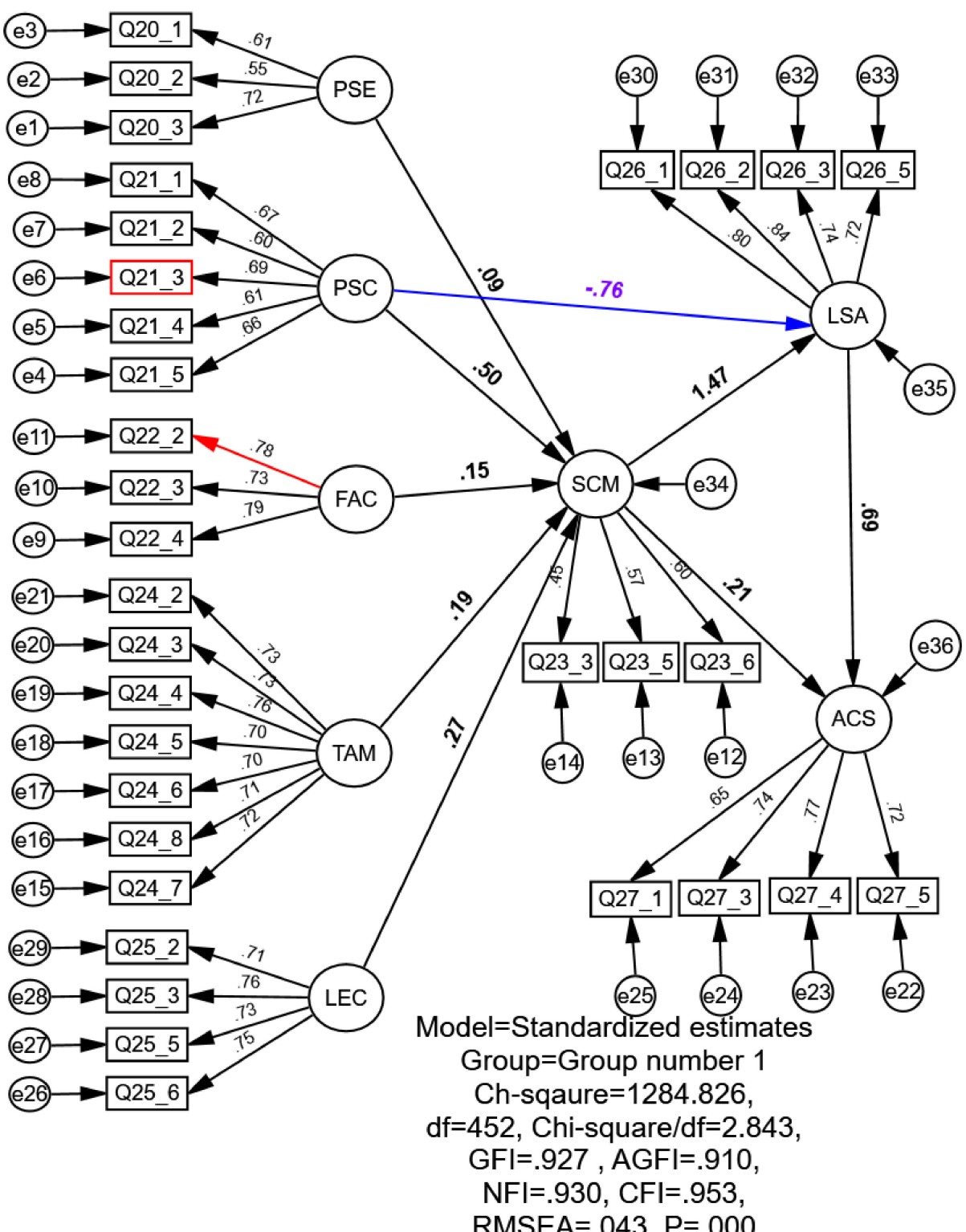

**Figure A15.** The revised model of mediating test: PSC → SCM → LSA (the blue line color is suggested relation for mediating effect).

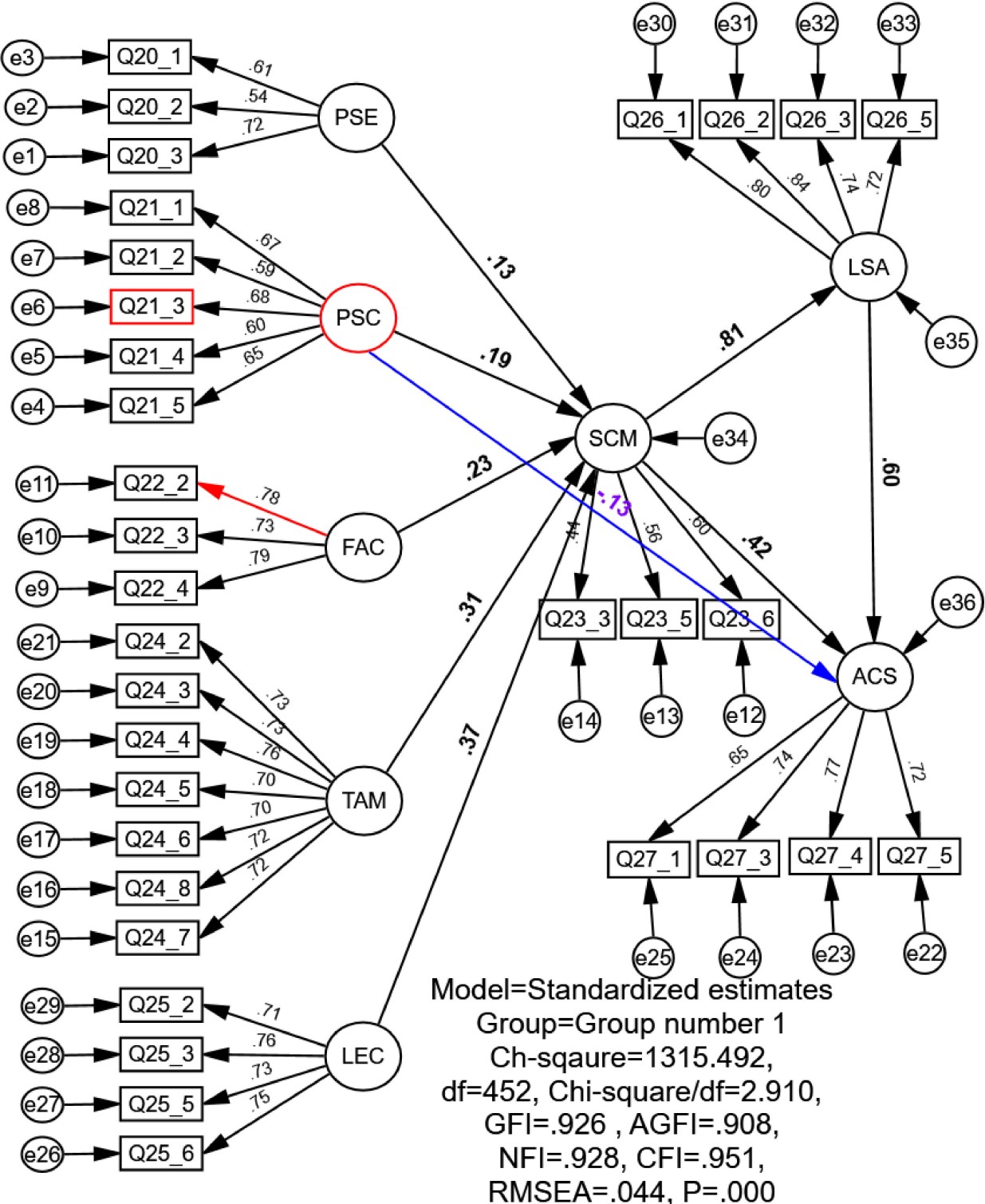

**Figure A16.** The revised model of mediating test: PSC → SCM → ACS (the blue line color is suggested relation for mediating effect).

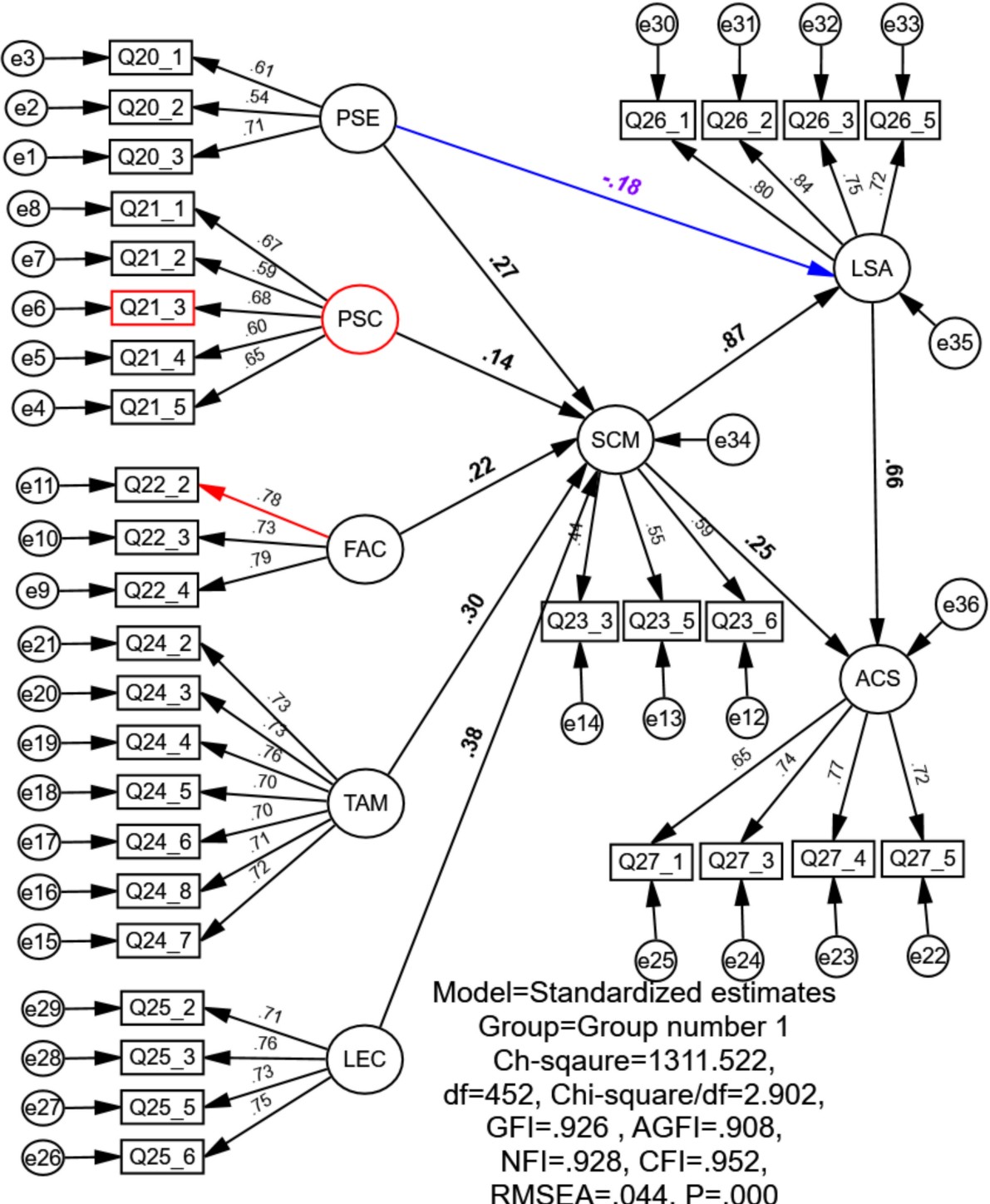

**Figure A17.** The revised model of mediating test: PSE → SCM → LSA (the blue line color is suggested relation for mediating effect).

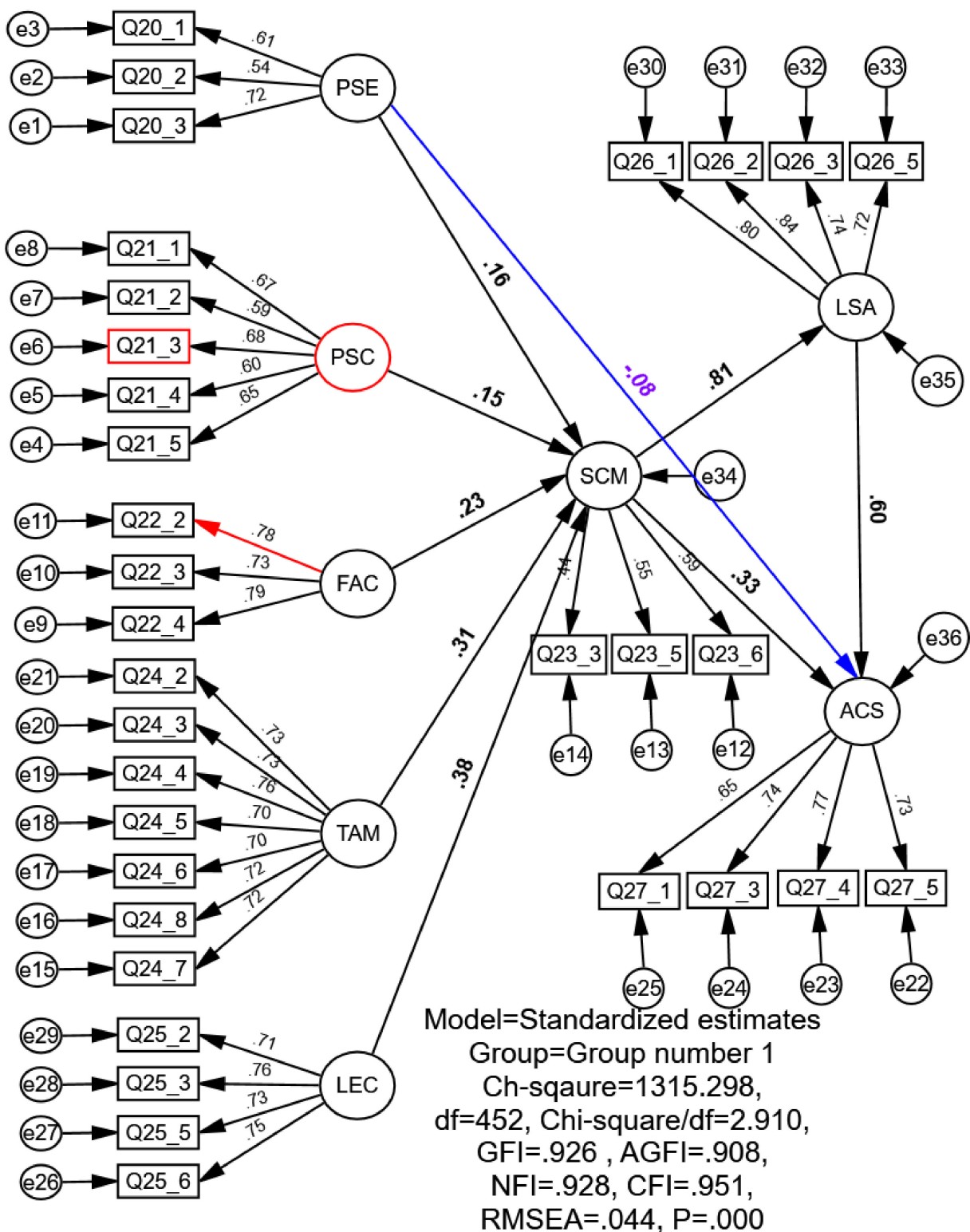

**Figure A18.** The revised model of mediating test: PSE → SCM → ACS (the blue line color is suggested relation for mediating effect).

## Appendix B

**Table A5.** List of key informants, focus group discussion and consultative meeting.

| Code | Position | Date |
|---|---|---|
| Pers. Comm. KII–1 | Department of Planning, Ministry of Education Youth and Sport (MoYES) | January 2022 |
| Pers. Comm. KII–2 | Department of Policy, Ministry of Education Youth and Sport (MoYES) | December 2021 |
| Pers. Comm. KII–3 | Department of Higher Education, Ministry of Education Youth and Sport (MoYES) | January 2022 |
| Pers. Comm. KII–4 | Vice–Rector in charge of students and ICT, Royal University of Phnom Penh (RUPP) | January 2022 |
| Pers. Comm. KII–5 | Lecturers, Royal University of Phnom Penh (RUPP) | December 2021 |
| Pers. Comm. KII–6 | Students, Royal University of Phnom Penh (RUPP) | December 2021 |
| Pers. Comm. FDG–1 | Focus Group Discussion among students, Royal University of Phnom Penh (RUPP) | December 2021 |
| Pers. Comm. CM–1 | Consultative Meeting among lecturers, Royal University of Phnom Penh (RUPP) | December 2021 |

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
