# Peer review of "The Antecedents and Consequences of Study Commitment to Online Learning at Higher Education Institutions (HEIs) in Cambodia"

_sustainability, doi:10.3390/su14063184_

Round 1

Reviewer 1 Report

In this paper cohort of 1002 undergraduate students in Cambodia are considered and the adaptation of online teaching and learning, consequences and antecedents of students’ study commitment to online learning, and factors influencing students’ willingness to participate in online learning during the post-COVID-19 pandemic is investigated. Finding is presented in the following categories:

  • two stages of online teaching and learning processes were adopted: individually-managed and institutionally-managed processes;
  • The students’ study commitment played an active role in improving their learning satisfaction. Meanwhile, academic support is one of the most outstanding factors influencing students’ online learning
  • In the post-COVID pandemic, 81.4% of undergraduate students did not propose to continue online learning. The survey confirms that online learning significantly reduced their academic performance, and 62.3% claimed online teaching negatively affected their studies.

The paper is very well-written and it is easy to read it. As we are coming out of the pandemic gradually, this study is very interesting and very useful to adopt/make changes to teaching and learning environment and policies in higher education. The paper is well-structured and I would recommend it for publication. However, the authors might want to consider the following comments to improve the work.

Please see my comments as following:

1- Data in some Figures are not very clear. For example, the weight on Figure 2.

2- In the discussion section, there is a need to also look at some the waste in both online and face to face learning and teaching process. The waste is briefly discussed is Section 5.1 “Lecturers often waste much time repeating their lecturers during poor internet services….”. In some recent publications Lean methodology is used to identify the waste in face to face and online teaching. I would recommend the authors refer to these works as well to make the discussion section more comprehensive. Example of these publications are:

  • “The effect of teaching–learning environments on student’s engagement with lean mindset”
  • “Lean management and sustainable practices in higher education institutions of Brazil and Portugal: a cross country perspective”
  • “The new role of teachers after the covid19 pandemic and the application of the lean concept in education.”

Author Response

Dear Reviewer

Thank for your kind comments and advice. We have worked on your comments seriously.

With best regards

Serey

Reviewer 2 Report

Dear colleagues,

The study is of high interest and importance at the period when many HEIs (Higher Education Institutions) administrations have got the illusion that online learning is a good solution to decrease costs with the maintenance of the quality. The maintenance of the quality requires to increase costs to support teachers and students.

The study results demonstrate the efforts and specific conditions determining the efficiency of online learning for students results and commitment, including, the interest to continue to study in online form.

Research design, chosen methodology and cohesion between results and conclusions is correct.

I would recommend checking the correction of the whole Abstract from the point of view of the editing (the content is correct, but some sentences could be formulated more clearly). 

Some small corrections are necessary:

- in the first line of Abstract: A rapid spread of the COVI-19 - you mean, COVID-19, I guess;

- line 19 - it seems, it would be added a link between "cohort" and "it was found", maybe, "In the study of the cohort..."

- line 82 misses a comma in the Ministry title (Cambodia’s Ministry of Education Youth and Sport - comma before Youth?) https://moeys.gov.kh/en/ - the official site puts comma.

- line 117 (and the same for line 175) - should the last sentence finish with words "the post-COVID-19 pandemic period." - ? (instead of "the post-COVID-19 pandemic.")

- lines 159-162 - the quatation in marks should be followed with a reference

- line 195 - the RUPP is one university, it is HEI (instead of HEIs)

- lines 381 and  947 - please, check the names of platforms - among Zoom, Google Meet, Massager, and Skype - the zoom, GM and skype are wellknown, but I had never heard of a Massager. (maybe, messengers? - comparing with the line 395 and 397).

- line 576 - maybe, "devices" instead of "divides"? (..."using platforms and electronic divides".)

- lines 663-668 - it would be interesting to go deeper to understand the reasons of this low efficiency in University support - e.g., which means and communication tools were used, what kinds of concerns, etc. - it could be subject for further research.

- lines 853-868 - the 7th point of Discussions - relates to the analysis of values that motivate students to studies. Again, this remark could be an advice for further research.

Good luck for continuation !

Author Response

(The authors gave the same response as above.)

Reviewer 3 Report

Abstract
The research method, process, tool, and sampling method should be stated in the abstract.

Keywords
No need to put "SEM" and "CFA" as keywords due to the keywords being to much and this paper was not discussing the statistic analysis methodology.

1. Introduction
The author should introduce the current situation of higher education in Cambodia, e.g. the policy of higher education toward online learning, the number of undergraduates, the number of online courses, the scale of the universities, etc in Cambodia.

2. Conceptualizing study commitment of online learning during COVID-19 pandemic
The spelling of the word "Accademic" in figure 1 should be checked.
It is not clear of relationships between factors in the research model. The association between factors should be supported by related literature one by one.

3. Materials and Methods
The pilot study should be conducted to test the reliability and validity of the research tool.
Please check and answer the Common Method Variance (CMV) problem in the research model.
Please show the mediating effects check result for LSA (learning satisfaction).

4. Results
The criteria in the Table 2 "Standardized Loading <0.60"? "AVE >0.50"? "CR >0.70"? Please double confirm the correct criteria.

5. Discussion
The discussion should be based on research hypotheses and analysis results.
The serial number of the subtitle should be double-checked, "5.1" then "4.2", "4.3"?

The parameters should be listed in Appendix B.7.

Author Response

(The authors gave the same response as above.)
